# A Theory of Learning with Competing Objectives and User Feedback

## Abstract

Large-scale deployed learning systems are often evaluated along multiple objectives or criteria. But, how can we learn or optimize such complex systems, with potentially conflicting or even incompatible objectives? How can we improve the system when user feedback becomes available, feedback possibly alerting to issues not previously optimized for by the system? We present a new theoretical model for learning and optimizing such complex systems. Rather than committing to a static or pre-defined tradeoff for the multiple objectives, our model is guided by the feedback received, which is used to update its internal state. Our model supports multiple objectives that can be of very general form and takes into account their potential incompatibilities. We consider both a stochastic and an adversarial setting. In the stochastic setting, we show that our framework can be naturally cast as a Markov Decision Process with stochastic losses, for which we give efficient vanishing regret algorithmic solutions. In the adversarial setting, we design efficient algorithms with competitive ratio guarantees. We also report the results of experiments with our stochastic algorithms validating their effectiveness.

## 1 Introduction

Learning algorithms trained on large amounts of data are increasingly adopted in a variety of applications and form the engine driving complex large-scale systems such as e-commerce platforms, online advertising auctions and recommender systems. Their system designer must take into account multiple metrics when optimizing them [Kaminskas and Bridge, 2016, Masthoff, 2011, Lin et al., 2019]. As an example, consider the case of a recommendation system for recipes, videos or fashion. There is no single metric that defines what a good recommendation engine should do. One needs to carefully take into consideration metrics measuring the quality of recommendations provided to end-users, their relevance and utility, the long-term growth of the content creators, and the overall revenue generated for the hosting platform. Furthermore, it is crucial to consider the risk of bias in these systems [Speicher et al., 2018, Xiao et al., 2017, Holstein et al., 2019]. Hence, additional metrics may need to be incorporated, such as performance across demographic groups, geographical locations or other identity terms. This can easily lead to hundreds of metrics that need to be simultaneously optimized for user satisfaction.

Further complicating the above scenario is the fact that often the multiple metrics considered are incompatible and inherently in conflict with each other [Kleinberg et al., 2017, Sener and Koltun, 2018, Jin, 2006]. For instance, in the context of a recommendation system, there is a tension between maximizing revenue via ad placements and maximizing end-user "happiness". Another tension may be between maximizing quality versus diversity of recommendations. In many cases, resolving such conflicts may force the designer to make hard choices among notions that seem perfectly reasonable in isolation, weighing in current use-patterns, wins and losses. An illuminating example is the analysis of the COMPAS tool for predicting recidivism by Angwin et al. [2019]. The authors showed that,

among black defendants who do not recidivate, the tool predicted incorrectly at twice the rate than it did for white defendants who did not recidivate, i.e., the tool was unfair according to the *false positive rate* metric. The creator of the tool, Northpointe, responded by demonstrating that the tool was fair according to other natural measures such as AUC (Area Under the ROC Curve), for which each group had similar values. Later work showed that this tension is inherent and that it is often impossible to simultaneously satisfy multiple seemingly natural criteria [Kleinberg et al., 2017] (see also Feller et al. [2016]).

The above discussion raises the question of how one should define the optimal trade-off among multiple conflicting metrics to optimize for user satisfaction. A natural approach is to define the trade-offs in a static manner, either by using domain knowledge and human expertise, or by analyzing past historical data. Another line of work studies optimization in the presence of multiple objectives by designing algorithms that compete with *any* linear combination of the objectives [Mohri et al., 2019, Cortes et al., 2020] or by designing pareto-optimal solutions [Sener and Koltun, 2018, Shah and Ghahramani, 2016]. However, these solutions may be sub-optimal for the richer situation where user feedback is available. While algorithms tailored to a specific metric or a combination of metrics would be effective at first, experience shows that they become non pertinent over time: once a system is deployed and it interacts with its end-users, inefficiencies in the system design emerge, as evident via the user feedback, which in turn could lead one to prefer metrics originally not accounted for [Liu et al., 2018]. Motivated by the above, in this work, we present a theoretical data-driven model for optimizing multiple conflicting metrics by taking into account the user feedback. Our proposed framework allows for the design of algorithmic solutions with strong theoretical guarantees.

In the context of a recommender system, user initiated feedback may be a "dislike", "too spicy", or "age inappropriate" [Chen and Pu, 2012], but feedback may also be indirectly observed by e.g. high abandonment rates or low click-through rates. Going from complaints to actionable solutions involves many steps. First, the complaints are analyzed, typically by human specialists, and attributed to a set of predefined criteria, such as low accuracy of classifiers, false positive rates or AUC scores. Each complaint could trigger several criteria and a human specialist can monitor the aggregate performance on each criterion. Since criteria are often incompatible, based on the analysis of the complaints and their affect on the criteria, a decision is made to allocate resources to improve a subset of the criteria and this process repeats [Yu et al., 2020]. While human involvement is crucial in the above process for both analyzing complaints and trading off metrics, a large portion of the above process could be made algorithmic and automated.

In practice, the problem of multiple conflicting metrics may emerge, even when a single fixed criterion is adopted [Klinkman et al., 1998, Buolamwini and Gebru, 2018]. As an example, consider again a recommendation system for videos. Let us assume that a system designer has opted for the false positive rate and the false negative rate to measure the performance of the system. The overall false positive (FP) rate or the false negative (FN) rate is rarely a good indicator of performance, particularly from an algorithmic bias point of view. Instead, the system designer would wish to monitor and optimize the FP/FN rates across different slices of the data, such as "sports", "religion", "LGBTQ issues" videos, or videos originating from different geographic locations, or a combination of them. This could easily result in hundreds of relevant slices of the data, where each can be viewed as a separate metric. As discussed before, these slices will often admit mutual incompatibilities [Kleinberg et al., 2017, Feller et al., 2016]. Thus, a user feedback data-driven method is needed to make the optimal choice. Our main contribution is precisely a data-driven model and algorithms for that purpose. Not only is our proposed framework grounded in theory, it can also be effectively realized in practice as we will show later.

Our model assumes predetermined costs for user complaints along the multiple metrics. The difficulty in optimizing for user happiness arises from the fact that the nature and volume of the complaints depend on the state of the model. Of course if no complaints is received, an optimal state has been reached, but most often complaints will arise. Fixing the model to optimize for this set of complaints will most likely spur a different set of complaints, etc. Only by visiting all incompatible states of the model and observing the associated complaint set would one be able to fully optimize the model. Such an exhaustive search is prohibitive from both a time and development perspective. This paper presents a model that effectively reaches a beneficial state and provides performance guarantees.

The rest of the paper is organized as follows. In Section 2, we define our model. In the stochastic setting (Section 3), we show that our framework can be cast as a Markov Decision Process with

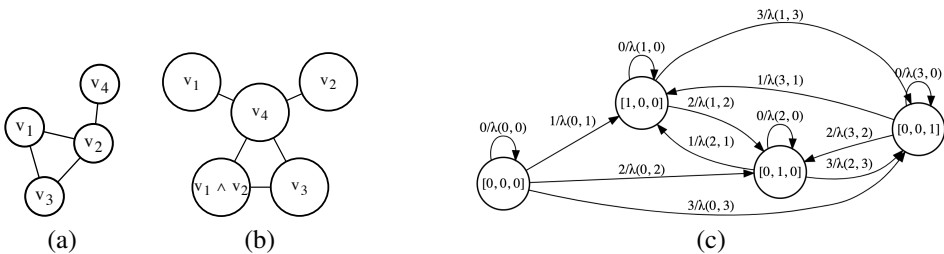

Figure 1: **(a)** Illustration of constraints graph $\mathcal{G}$. $v_1, v_2, v_3, v_4$ represent 4 different criteria. **(b)** More generally, each vertex can represent a joint criterion, for example $v_1 \wedge v_2$. This helps specify joint constraints such as the following: $v_1$, $v_2$, and $v_3$ cannot be simultaneously satisfied. **(c)** Illustration of the MDP for a fully connected incompatibility graph $\mathcal{G}$ over three criteria. The state set is $\mathcal{S} = \{\mathbf{0} = [0,0,0], \mathbf{1} = [1,0,0], \mathbf{2} = [0,1,0], \mathbf{3} = [0,0,1]\}$, the action set $\mathcal{A} = \{0,1,2,3\}$. Each transition is labeled with $a/\lambda(s,a)$, where $a$ is the action taken from $s$ and where $\lambda(s,a)$ is the total loss incurred as a result.

stochastic losses, for which we give efficient vanishing regret algorithmic solutions. In the adversarial setting (Section 4), we give algorithms with competitive ratio guarantees. Appendix D demonstrates how our framework can be realized in practice and reports the results of experiments with our algorithms in the stochastic setting that demonstrate their effectiveness and the applicability of our model. We further discuss our modeling assumptions and extensions in Appendix B, and finally, discussion of related work and proofs of theorems can also be found in the appendix.

## 2 Conflict resolution model

We consider optimization in the presence of multiple criteria, where not all criteria can be satisfied simultaneously. The constraints are specified by an undirected graph $\mathcal{G} = (V, E)$, where each vertex represents a criterion and where an edge between vertices $v_i$ and $v_j$ indicates that criteria $v_i$ and $v_j$ cannot be simultaneously satisfied. We denote by $V = \{v_1, \ldots, v_k\}$ the set of $k$ criteria considered. Figure 1 illustrates these definitions. Note that vertices may represent joint criteria as in Figure 1(b).

We consider a machine learning system that evolves over a sequence of time steps in the presence of the criteria represented by the graph $\mathcal{G}$. At each time step $t$, the system is in some state $s_t$ characterized by its performance on all criteria in $V$. Note, a state is distinct from a vertex of $\mathcal{G}$. The system then receives a new batch of feedback that depend on its current state and incurs a loss. The objective of the algorithm is to minimize the total cost incurred over a period of time, which includes the total loss accrued, as well as the total cost of fixing various criteria over that period. We envision that the algorithm is solving a constraint optimization problem with the criteria as constraints.

The assignment of a complaint to one or more criteria can be achieved by human analysis or via a multi-class multi-label classifier trained on past data and making use of known classifiers for specific criteria. Even when a complaint is related to a single criterion, we do not simply advocate taking that raw feedback as the ground truth. We discuss the risks associated with doing so in Section 3 and Appendix E, in the context of the COMPAS example. To further improve and maintain the accuracy of this multi-class multi-label classifier, in practice, there may be ongoing data labeling and assistance by expert auditors analyzing complaints. Note that not all complaints received by the system are relevant and the classifier, or a human in the loop, may decide to not assign a complaint to any criterion. This also helps protect the system against potential attacks by coordinated users. Recent work on interactive models for ML fairness has studied this for specific metrics and auditor behavior [Bechavod et al., 2020].

**Loss**. As a result of the complaints, the system incurs a loss and responds by changing its state. The definition of the loss, which depends on the criteria affected by the complaints is critical, a poor choice can yield a so-called *loudest voice* effect (see discussion in Section 3). The notion of complaints and the associated loss may seem abstract at this point. In Appendix D, we demonstrate how our model can be applied in practice.

**Graph and criteria**. The assumption that the graph $\mathcal{G}$ is known a priori may seem restrictive. However, in many settings, graph $\mathcal{G}$ can be derived from analyzing past complaints and by measuring how fixing one criterion affects the performance on others. For instance, in the recommendation

system example discussed above, where each metric corresponds to the false positive rate on a different slice of the data, one can easily use past data to see how optimizing the false positive rate on one slice affects the other and get the graph of incompatibilities. See the experiments in Section D for a more concrete example. Our model also provides the flexibility of accounting for incompatibilities among criteria such as those discussed by Kleinberg et al. [2017] and Feller et al. [2016]. This can be achieved by augmenting the graph with vertices representing joint criteria as in Figure 1(b). The graph stipulates in particular that $v_1$, $v_2$ and $v_3$ cannot be all simultaneously satisfied.

**States**. For our theoretical and algorithmic analysis, we will adopt the following simplifying assumptions and will later discuss their extensions or relaxation in Section 3. We assume that each criterion can only be in one of two states: *fixed*, meaning that criterion $v_i$ is met or is not violated, or *unfixed*, meaning the opposite. Hence, the overall state of the system can be described by a $k$-dimensional Boolean vector. An action corresponds to fixing a particular criterion, or set of criteria, and moving to a different vertex $v_i$ in the graph. *Fixing* the criteria associated to $v_i$ entails an algorithmic and resource allocation cost that we denote by $c_i$. Initially, all criteria are unfixed. At each time step, a conflict resolution system or algorithm selects some action, which may be to fix an unfixed vertex $v_i$, thereby incurring the cost $c_i$ and *unfixing* any vertex adjacent to $v_i$, or the algorithm may select the null action, not to fix or unfix any vertex and wait to collect more data. Note that the incompatibilities in our model defined via edges in the graph are data agnostic. In practice, it is possible that two generally incompatible criteria can be simultaneously satisfied for a given dataset, say via incorporating a slack. This is a direction for future work.

**Fixing costs**. The fixing cost can be estimated from past experience. In the absence of any prior information, one could assume a unit fixing cost for all criteria. We deliberately avoid making specific choices. This gives us flexibility in dealing with different types of metrics in a unified manner.

# 3 Stochastic setting

We first detail a stochastic setting of our model that can be described in terms of a Markov Decision Process (MDP). Next, we present algorithms with strong regret guarantees.

**Description.** The distribution of complaints received by the system is a function of its current *state*, that is the current set of fixed or unfixed criteria $v_i$. Thus, we consider an MDP with a state space $\mathcal{S} \subseteq \{0,1\}^k$ representing the set of bit vectors for criteria: a state $s \in \{0,1\}^k$ is defined by $s(i) = 0$ when criterion $v_i$ is unfixed and $s(i) = 1$ when it is fixed. By definition of the incompatibility graph $\mathcal{G}$, $s$ is a valid state if and only if the set of fixed criteria at $s$ is an independent set of $\mathcal{G}$.

When in state $s \in \mathcal{S}$, the system incurs a loss $\ell_i^s$ due to complaints related to criterion $i \in [k]$. Loss $\ell_i^s$ is a random variable assumed to take values in $[0, B]$ with mean $\mu_i^s$. We do not assume independence across criteria, i.e., $\ell_i^s$ and $\ell_j^s$ may be dependent for a given state $s$. The action set is $\mathcal{A} = \{0, 1, \ldots, k\}$. A non-zero action $i$ corresponds to fixing criterion $i$. Action 0 is the null action, that is no criterion is fixed. Transitions are deterministic: given state $s$ and action $i \in \mathcal{A}$, the next state is $s$ if $i = 0$ since the fixed-unfixed bits for criteria are unchanged; otherwise, for $i \neq 0$ the next state is the state $s'$ that only differs from $s$ by $s'(i) = 1$ and (possibly) $s'(j) = 0$ for all $j \in N(i)$, where $N(i)$ is the neighbors of $v_i$ in $\mathcal{G}$, since neighbors of $i$ must be unfixed once $i$ is fixed.

Each action $a = i$ admits a fixing cost $c_i$. The cost for unfixing, as well as the null action, is zero. The loss incurred when taking action $a$ at state $s$ is the sum of the fixing cost $c_a$ and the complaint losses at the (possibly) next state $s'$: $\lambda(s, a) = c_a + \sum_{i=1}^k \ell_i^{s'}$. The expected loss of transition $(s, a, s')$ is:

$$\mathbb{E}\left[ c_a + \sum_{i=1}^k \ell_i^{s'} \right] = c_a + \sum_{i=1}^k \mu_i^{s'}. \tag{1}$$

Note, $c_a$ and the losses $\ell_i^{s'}$ are observed by the algorithm, but the mean values $\mu_i^{s'}$ are unknown. To keep the formalism simple we assume that the cost $c_a$ of taking an action $a$ is independent of the current state $s$. Figure 1 (c) illustrates our stochastic model for three mutually incompatible criteria. The notion of each metric in a binary state is a simplifying modeling assumption for our theoretical investigation. We discuss this more at the end of the Section.

**Correlation sets.** In practice, the distribution of complaints related to a criterion $v_i$ at two different states may be related. To capture these correlations in a general way, we

assume that a collection $\mathcal{C} = \{\mathcal{C}_1, \mathcal{C}_2, \ldots, \mathcal{C}_n\}$ of *correlation sets* is given, where each $\mathcal{C}_j$ is a subset of the $k$ criteria and has size at most $m$. By allowing correlation sets of varying sizes, we can capture a range of dependencies that may exist between different criteria. These dependencies affect the loss observed by the algorithm at each time.

We assume that at a given state $s$, each set $\mathcal{C}_j$ generates losses with mean value $\theta_j^s$ per vertex, and that if two states $s$ and $s'$ admit the same configuration for the vertices in $\mathcal{C}_j$, then they share the same parameter $\theta_j^s = \theta_j^{s'}$. Given a criterion $i$ and a state $s$, we assume that the loss incurred by criterion $i$ equals the sum of the individual losses due to each correlation set $\mathcal{C}_j$ that contains $i$. Thus, $\mu_i^s$ can be expressed as follows: $\mu_i^s = \sum_{j=1}^{n} \theta_j^s \mathbb{1}(i \in \mathcal{C}_j)$. If a criteria is not correlated with any other vertex, we add to $\mathcal{C}$ a correlation set of size one for that criterion. See Figure 2 for an illustration. For each $j \in [n]$, there are at most $2^m$ configurations for the vertices of $\mathcal{C}_j$ in a state $s$, hence there are at most $2^m n$ distinct parameters $\theta_j^s$. Let $\boldsymbol{\theta}$ denote the vector of all distinct parameters $\theta_j^s$. Our MDP model can then be denoted $\text{MDP}(\mathcal{S}, \mathcal{A}, \mathcal{C}, \boldsymbol{\theta})$.

$$
\begin{aligned}
\mu_1 &= \theta_1^s \\
\mu_2 &= \theta_1^s + \theta_3^s \\
\mu_3 &= \theta_3^s \\
\mu_4 &= \theta_2^s
\end{aligned}
$$

Figure 2: Example of correlation sets and associated losses for a graph with four criteria.

**Algorithm.** We consider an online algorithm that at time $t$ takes action $a_t$ from state $s_t$ and reaches state $s_{t+1}$, starting from the initial state $(0, \ldots, 0)$. The objective of an algorithm can be formulated as that of learning a policy, that is a mapping $\pi \colon \mathcal{S} \to \mathcal{A}$, with a value close to that of the optimal. We are mainly interested in the cumulative loss of the algorithm over the course of $T$ interactions with the environment. The goal is to minimize the pseudo-regret:

$$
\text{Reg}(\mathcal{A}) = \sum_{t=1}^{T} \mathbb{E}\left[\lambda_t(s_t, a_t)\right] - \sum_{t=1}^{T} \mathbb{E}\left[\lambda_t(s_t^{\pi^*}, \pi^*(s_t^{\pi^*}))\right], \tag{2}
$$

where $\lambda_t(s, a)$ is the total loss incurred by taking action $a$ at state $s$ at time $t$, $s_1 = (0, \ldots, 0)$ and $\pi^*$ is the optimal policy. Note, $\lambda_t$ is only a function of the current state and the action taken. The expectation is over the random generation of the complaint losses. Given the correlation sets and the parameter $\boldsymbol{\theta}$, the optimal policy $\pi^*$ corresponds to moving from the initial state $(0, \ldots, 0)$ to the state $s^* \in \mathcal{S}$ with the most favorable distribution and remaining at $s^*$ forever. We define by $g(s)$ the expected (per time step) loss incurred by staying in state $s$, that is, $g(s) := \sum_{i=1}^{k} \mu_i^s$. The optimal state $s^*$ is then defined as follows:

$$
s^* = \operatorname*{argmin}_{s \in \mathcal{S}} g(s). \tag{3}
$$

Note, in this definition of $s^*$, we disregard the one-time cost of moving to a state from the initial state, since in the long run the expected cost incurred by staying at a given state governs the choice of the optimal state. We will assume that we have access to an oracle that can solve the above offline optimization problem. This is a standard assumption in the theory of online learning and MDPs. Since our problem can be seen as that of learning with a deterministic MDP with stochastic losses, we could adopt an existing algorithm for that problem [Jaksch et al., 2010]. However, the running-time of such algorithms would directly depend on the size of the state space $\mathcal{S}$, which here is exponential in $k$, and that of the action set $\mathcal{A}$. Furthermore, the regret guarantees of these algorithms would also depend on $|\mathcal{S}||\mathcal{A}|$.

**Case $m = 2$.** We first consider a simpler setting where correlation sets are defined on subsets of size at most two. This setting also captures an important case where fixing a particular criterion affects the complaints of its neighbors. The algorithmic challenge we face here is to avoid exploring the exponentially many states in the MDP. Instead, we will design an algorithm that spends an initial exploration phase by visiting a specific subset of states of size at most $4n$. This subset denoted by $\mathcal{K}$, that we call a *cover* of $\mathcal{C}$ will help the algorithm estimate the expected loss of any state in the MDP given the estimates of losses for states in the cover. We next formally define the cover.

For two criteria $i, j$ and $b \in \{0, 1\}$, we say that $(i, j, b)$ is a *dichotomy* if there exist two states $s, s' \in \mathcal{S}$ such that: (1) $s(j) = 0$ and $s'(j) = 1$, and (2) $s(i) = s'(i) = b$. We call the two states $s, s'$ an $(i, j, b)$-pair. Note that if an edge $(v_i, v_j)$ is present in $\mathcal{G}$, then $(i, j, 1)$ cannot be a dichotomy, since criteria $i$ and $j$ cannot be fixed simultaneously. A cover $\mathcal{K}$ of $\mathcal{C}$ is simply a subset of the states in the MDP that contains an $(i, j, b)$-pair for every $\{i, j\} \in \mathcal{C}$ and valid dichotomy $(i, j, b)$.

Furthermore, for every singleton set $\{i\}$ in $\mathcal{C}$, $\mathcal{K}$ contains states $s, s'$ such that $s(i) = 0, s'(i) = 1$ and $s(j) = s'(j)$ for all $j \neq i$. Note that we only need the cover to contain an $(i, j, b)$-pair if $\{i, j\}$ is

a correlation set. Hence, it is easy to see that when $m = 2$, there is always a cover of size at most $4n$. We then have the following guarantee.

**Theorem 1.** *Consider an MDP$(\mathcal{S}, \mathcal{A}, \mathcal{C}, \boldsymbol{\theta})$ with losses in $[0, B]$, maximum fixing cost $c$, and correlations sets of size at most $m = 2$. Let $\mathcal{K}$ be a cover of $\mathcal{C}$ of size $r \leq 4n$, then, the algorithm of Figure 3 (see Appendix B) achieves a pseudo-regret bounded by $O(kr^{1/3}(c + B)(\log rkT)^{1/3}T^{2/3})$. Furthermore, given access to an oracle for (3), the algorithm runs in time polynomial in $k$ and $n = |\mathcal{C}|$.*

There is a natural extension to arbitrary correlation sets via extending the notion of a dichotomy and a cover (Algorithm in Figure 4, Appendix B). Our algorithms are also scalable. During step 1 they only explore the states in the cover $\mathcal{K}$ that could be much smaller than the full state space.

**Beyond $T^{\frac{2}{3}}$ regret.** Next, we present algorithms that achieve $\tilde{O}(\sqrt{T})$ regret. In particular for the case of $m = 1$, we have the following guarantee.

**Theorem 2.** *Consider MDP$(\mathcal{S}, \mathcal{A}, \mathcal{C}, \boldsymbol{\theta})$ with losses in $[0, B]$ and maximum fixing cost $c$. Given correlations sets $\mathcal{C}$ of size one, the algorithm of Figure 5 (see Appendix B.2) achieves a pseudo-regret bounded by $O(k^2(c + B)^2\sqrt{T}\log T)$. Furthermore, given access to an oracle for (3), the algorithm runs in time polynomial in $k$.*

The theorem above can also be extended to higher values of $m$ (see Figure 6 in Appendix B.2).

## 4 Adversarial setting

We also study a setting with no distributional assumptions about the arrival of complaints. We consider an adversarial model where, at each time step, multiple complaints arrive for the vertices in $\mathcal{G}$. Initially all the vertices in $\mathcal{G}$ are in an unfixed state and each vertex has a fixing cost of $c_i$. Each time, the algorithm can decide to fix a particular vertex, and as a result its neighbors get unfixed. At time step $t$, if criterion $v_i$ is unfixed, then the algorithm incurs a loss of $\ell_{i(t)}$ (which depends on the current state of the system), otherwise the algorithm incurs no loss. For an algorithm $\mathcal{A}$, during $T$ time steps, the total loss is

$$\text{Loss}(\mathcal{A}) = \sum_{i=1}^{k}\sum_{t=1}^{T} \ell_{i(t)} \cdot \mathbb{1}(s_t(i) = 0) + \sum_{i=1}^{k}\sum_{t=2}^{T} c_i \cdot \mathbb{1}(s_{t-1}(i) = 0, s_t(i) = 1). \tag{4}$$

Let OPT be the algorithm that, given the entire loss sequence in advance, makes the decisions to fix vertices. We define the *competitive ratio* [Borodin and El-Yaniv, 1998] of $\mathcal{A}$ to be the maximum of $\text{Loss}(\mathcal{A})/\text{Loss}(\text{OPT})$ over all possible complaint sequences. Our main result is stated below.

**Theorem 3.** *Let $\mathcal{G}$ be a graph with fixing costs at least one. There is a polynomial-time algorithm with a competitive ratio of at most $2B + 4$ on any sequence of complaints with loss values in $[0, B]$.*

Our algorithm for this setting is provided in Figure 7 in Appendix C.

## 5 Experiments

While our primary contribution is a theoretical framework and the design of near optimal algorithms, our proposed algorithms are indeed scalable and practical. We demonstrate this in Appendix D via experiments on both simulated and real world data.

## 6 Conclusion

We presented a new data-driven model of online optimization from user feedback in the presence of multiple criteria, with algorithms benefiting from theoretical guarantees both in the stochastic and the adversarial setting. We provided empirical evidence that our model can be effectively realized in practice. Several extensions are worth exploring in future work. These include fixing costs that can vary with time to capture varying algorithmic price and human effort cost. Similarly, the expected losses in our stochastic model could be time-dependent to express the growing cost of a criterion not being addressed.

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

# Table of Contents – Appendix

# A  Related work

There is extensive literature on optimizing multiple metrics or objectives under specific criteria. The recent works of Mohri et al. [2019], Cortes et al. [2020] consider optimizing in the presence of multiple base objectives. Given objectives $L_1, \ldots, L_i$ these works aim to design "agnostic" algorithms that can simultaneously compete with any linear or convex combination of the objectives. Another line of work considers design algorithms that can achieve the Pareto optimal solution [Jin and Sendhoff, 2008, Sener and Koltun, 2018, Shah and Ghahramani, 2016, Marler and Arora, 2004].

Another line of work considers optimizing multiple constraints (inspired by group fairness metrics) via constrained non-convex optimization [Agarwal et al., 2018, Cotter et al., 2018a, Thomas et al., 2019]. These publications either reduce the problem to that of cost-sensitive classification [Agarwal et al., 2018, Dwork et al., 2018] or replace the non-convex constraints by convex proxies and next optimize them via external or swap regret minimization algorithms [Cotter et al., 2018b,a].

There have also been studies of the inherent tension between satisfying multiple metrics. Kleinberg et al. [2017] and Feller et al. [2016] demonstrate that it is impossible to satisfy equal opportunity and calibration at the same time. Inspired from fairness applications the work of Menon and Williamson [2018] studies the tradeoff between accuracy and other metrics of interest such as false positive and false negative rates.

Since we are concerned with optimizing multiple metrics, it is natural to consider whether the problem can be framed via multi-task learning. However, there are certain crucial differences. In multi-task learning, the learner has access to data from multiple tasks and the goal is to jointly learn these tasks, which are assumed to be somewhat related or similar, to achieve a better generalization across all tasks. The online version of the problem admits many variants and with the aim of learning both a task similarity and predictors or only predictors when a task similarity is already supplied. The literature is indeed very rich.

In our setting, there is typically only one task (same label), but different loss functions. In lieu of a similarity between tasks, we have an incompatibility graph between losses. We consider user feedback which does not seem to have a direct counterpart in the multi-task setting. Furthermore, a different predictor is typically learned for each task, while this is not our setting. In most settings of multi-task online learning, the objective is in terms of an adversarial regret, while our MDP scenario is for a stochastic scenario. Hence, while there are some aspects that seem reminiscent of our scenario, the traditional multi-task learning scenario seems to be quite different from our considered setting.

Recent works have also studied the long-term impact of optimizing multiple conflicting criteria in settings with feedback mechanisms [Liu et al., 2018, Hashimoto et al., 2018, Mouzannar et al., 2019, Kannan et al., 2019]. Liu et al. [2018] show that, in certain situations, constrained loss minimization to equalize certain criteria could lead to further disparate impact on the end users in the long run. Hashimoto et al. [2018] proposed algorithms for minimizing such disparate impact in settings involving repeated loss minimization. More recently, Jabbari et al. [2017], Wen et al. [2019] study the problem of satisfying multiple constraints in reinforcement learning settings involving a Markov Decision Process. The authors in Jabbari et al. [2017] consider learning in an MDP where the criteria to be optimized require that the algorithm never takes an action $a$ over action $a'$ if the long-term reward is higher. It is clear to see that the optimal policy for the MDP indeed satisfies this property. Hence, there does exist a policy that satisfies the required criterion. However, the authors show that finding a near optimal policy while satisfying the criterion requires time exponential in the size of the state space.

Wen et al. [2019] consider other metrics such as demographic parity in the context of learning in MDPs. Doroudi et al. [2017] show that existing importance sampling methods for off-policy policy selection in reinforcement learning can lead to bad outcomes according to other natural criteria and present algorithms to mitigate this effect.

While our work also involves learning in a Markov Decision Process (MDP) and optimizing multiple criteria in the long term, the setup and the motivation are different. Unlike all the previous work mentioned, we do not commit to a fixed definition of quality or a metric, and allow for arbitrary criteria. Hence, states in our MDP correspond to the current configurations of different criteria. Rather than studying each metric in isolation, the objective of our work is to propose a data-driven model that can learn from feedback, a near-optimal configuration of the metrics to impose on the

system. To the best of our knowledge, ours is the first work to incorporate optimizing metrics of arbitrary types in an online setting. In this context, inspired by fairness applications, the recent work of Kearns et al. [2019] studies a specific combination of group and individual fairness metrics. The authors consider a setting where there is a distribution over individuals as well as a distribution over classification tasks. They consider algorithms for achieving *average* individual fairness, that is in expectation over classification tasks, the performance of the algorithm on a group fairness metric such as demographic parity should be the same for each individual.

An important aspect of our stochastic MDP-based model requires the ability to observe the losses associated with different criteria at each time. This relates to the problem of evaluating and monitoring the performance of the system according to different metrics from data. There has been work in recent years on developing auditing and monitoring approaches Bastani et al. [2019], Ghosh et al. [2020], Bellamy et al. [2018]. Furthermore, many metrics require access to both labeled data and to certain sensitive attribute information such as race or gender, for accurate evaluation. A recent line of work has studied this estimation problem when one has limited and/or noisy access to sensitive attribute information Gupta et al. [2018], Coston et al. [2019], Lamy et al. [2019], Wang et al. [2020]. Finally, we note that our model learns from feedback received as a form of complaints. These complaints are a result of a (potentially incorrect) decision made by an ML system. There has been recent work in developing counterfactual based explanations Tsirtsis and Gomez-Rodriguez [2020] for such decisions and exploring recourse strategies Gupta et al. [2019].

## B Stochastic setting

In this section we provide algorithms and their analysis for the stochastic setting as defined in Section 3. Recall from the setup in Section 3 that since our problem can be seen as that of learning with a deterministic MDP with stochastic losses, we could adopt an existing algorithm for that problem [Jaksch et al., 2010]. However, the running-time of such algorithms would directly depend on the size of the state space $\mathcal{S}$, which here is exponential in $k$, and that of the action set $\mathcal{A}$. Furthermore, the regret guarantees of these algorithms would also depend on $|\mathcal{S}||\mathcal{A}|$. Instead, by exploiting the structure of the MDP, we can design vanishing regret algorithms with a computational complexity that is only polynomial in $k$ and the number of parameters. We will assume access to an oracle that, given $\boldsymbol{\theta}$, can optimize (3). In Appendix B.3, we show how to approximately solve (3) for the case of $m = 1$, i.e., singleton correlation sets. In that case, the true parameters $\boldsymbol{\theta}$ can also be estimated efficiently (see Theorem 9).

**Case $m = 2$.** To illustrate the ideas behind our general algorithm, we first consider a simpler setting where correlation sets are defined on subsets of size at most two. We first recall the notion of a *cover* from Section 3.

For two criteria $i, j$ and $b \in \{0, 1\}$, we say that $(i, j, b)$ is a *dichotomy* if there exist two states $s, s' \in \mathcal{S}$ such that: (1) $s(j) = 0$ and $s'(j) = 1$, and (2) $s(i) = s'(i) = b$. We call the two states $s, s'$ an $(i, j, b)$-pair. Note that if an edge $(v_i, v_j)$ is present in $\mathcal{G}$, then $(i, j, 1)$ cannot be a dichotomy, since criteria $i$ and $j$ cannot be fixed simultaneously. A cover $\mathcal{K}$ of $\mathcal{C}$ is simply a subset of the states in the MDP that contains an $(i, j, b)$-pair for every $\{i, j\} \in \mathcal{C}$ and valid dichotomy $(i, j, b)$.

Furthermore, for every singleton set $\{i\}$ in $\mathcal{C}$, $\mathcal{K}$ contains states $s, s'$ such that $s(i) = 0, s'(i) = 1$ and $s(j) = s'(j)$ for all $j \neq i$. Note that we only need the cover to contain an $(i, j, b)$-pair if $\{i, j\}$ is a correlation set. Hence, it is easy to see that when $m = 2$, there is always a cover of size at most $4n$.

Next, we state our key result that estimating the loss values for the states in a cover is sufficient.

**Theorem 4.** *Let $\mathcal{K}$ be a cover for $\mathcal{C}$. For any state $s \in \mathcal{S}$ and any $i \in [k]$ with $s(i) = b$, we have:*

$$\mu_i^s = \mu_i^{s'} + \sum_{j=1}^{k} X_b^{i,j} \left[ \mathbb{1}(s(j) = 1) \, \mathbb{1}(s'(j) = 0) \right] - \sum_{j=1}^{k} X_b^{i,j} \left[ \mathbb{1}(s(j) = 0) \, \mathbb{1}(s'(j) = 1) \right], \quad (5)$$

*where $s'$ is any state in $\mathcal{K}$ with $s'(i) = b$, and for $\{i, j\} \in \mathcal{C}$, $X_b^{i,j} := \mu_i^{s_1} - \mu_i^{s_2}$ where $(s_1, s_2)$ is some $(i, j, b)$ pair. If $\{i, j\} \notin \mathcal{C}$, we define $X_b^{i,j}$ to be zero.*

*Proof.* Consider a correlation set $\{i, j\}$. The expected loss incurred by vertex $v_i$ or $v_j$ due to this set in any given state depends solely on the configuration of $v_i$ and $v_j$ in that state. Hence there are four parameters in the $\boldsymbol{\theta}$ vector corresponding to the correlation set $\{i, j\}$ and we denote them using $\gamma_{i,j}^{a,b}$, where $a, b \in \{0, 1\}$. Let $s, s'$ be an $(i, j, b)$ pair. When we switch from $s$ to $s'$ the only difference in the expected losses for vertex $i$ comes from the pair $(i, j)$. Hence we have

$$\mu_i^{s'} - \mu_i^s = \gamma_{i,j}^{b,1} - \gamma_{i,j}^{b,0} := X_b^{i,j}.$$

Hence, given the loss estimates for states in $\mathcal{K}$ we can estimate $X_b^{i,j}$ for each $i, j \in [k]$ and $b \in \{0, 1\}$. Next, given an arbitrary state $s$ with $s(i) = b$ let $s'' \in \mathcal{K}$ such that $s''(i) = b$. We have

$$
\begin{aligned}
\mu_i^s &= \mu_i^{s''} + \sum_{\substack{j:s(j)=0 \\ s''(j)=1}} (\gamma_{i,j}^{b,0} - \gamma_{i,j}^{b,1}) + \sum_{\substack{j:s(j)=1 \\ s''(j)=0}} (\gamma_{i,j}^{b,1} - \gamma_{i,j}^{b,0}) \\
&= \mu_i^{s''} + \sum_{\substack{j:s(j)=1, \\ s''(j)=0}} X_b^{i,j} - \sum_{\substack{j:s(j)=0, \\ s''(j)=1}} X_b^{i,j} \\
&= \mu_i^{s''} + \sum_{j=1}^{k} X_b^{i,j} \left[ \mathbb{1}(s(j) = 1) \, \mathbb{1}(s''(j) = 0) - \mathbb{1}(s(j) = 0) \, \mathbb{1}(s''(j) = 1) \right].
\end{aligned}
$$

$\square$

> **Input:** The graph $\mathcal{G}$, correlation sets $\mathcal{C}$, fixing costs $c_i$.
> 1. Pick a cover $\mathcal{K} = \{s_1, s_2, \ldots, s_r\}$ of $\mathcal{C}$.
> 2. Let $N = 10\frac{T^{2/3}(\log rkT)^{1/3}}{r^{2/3}}$.
> 3. For each state $s \in \mathcal{K}$ do:
>    - Move from current state to $s$ in at most $k$ time steps.
>    - Play action $a = 0$ in state $s$ for the next $N$ time steps to obtain an estimate $\widehat{\mu}_i^s$ for all $i \in [k]$.
> 4. Using the estimated losses for the states in $\mathcal{K}$ and Equation (5), run the oracle for the optimization (3) to obtain an approximately optimal state $\hat{s}$.
> 5. Move from current state to $\hat{s}$ and play action $a = 0$ from $\hat{s}$ for the remaining time steps.

Figure 3: Algorithm for $m = 2$ achieving $\tilde{O}(T^{2/3})$ pseudo-regret.

From the above theorem we have the following guarantee.

**Theorem 5.** *Consider an MDP$(\mathcal{S}, \mathcal{A}, \mathcal{C}, \boldsymbol{\theta})$ with losses in $[0, B]$, maximum fixing cost $c$, and correlations sets of size at most $m = 2$. Let $\mathcal{K}$ be a cover of $\mathcal{C}$ of size $r \leq 4n$, then, the algorithm of Figure 3 achieves a pseudo-regret bounded by $O(kr^{1/3}(c+B)(\log rkT)^{1/3}T^{2/3})$. Furthermore, given access to an oracle for (3), the algorithm runs in time polynomial in $k$ and $n = |\mathcal{C}|$.*

*Proof.* In each time step the maximum loss incurred by any criterion is bounded by $c + B$. Let $\{s_1, s_2, \ldots, s_r\}$ be the states in $\mathcal{K}$. During the exploration phase the algorithm stays in each state for $N$ time steps and incurs a total loss bounded by $kNr(c + B)$. During the exploration phase the algorithm moves from one state to another in at most $k$ steps and incurs a total additional loss of at most $rk^2(c + B)$. At any given state $s \in \mathcal{K}$ and vertex $v_i$, after $N$ time steps we will, with probability at least $1 - \delta$, an estimate of $\mu_i^s$ up to an accuracy of $2B\sqrt{\frac{\log 1/\delta}{N}}$. Setting $\delta = 1/(rkT^4)$ and using union bound, we have that at the end of the exploration phase, with probability at least $1 - \frac{1}{T^4}$, the algorithm will have estimate $\hat{\mu}_i^s$ for all $s \in \mathcal{K}$ and $i \in [k]$ such that

$$\hat{\mu}_i^s - \mu_i^s \leq 4B\sqrt{\frac{\log rkT}{N}}. \tag{6}$$

Hence during the exploitation phase, with high probability, the algorithm will have the estimate for the expected loss of each state in $\mathcal{S}$, i.e., $\sum_i \mu_i^s$ up to an error of $4kB\sqrt{\frac{\log rkT}{N}}$. Combining the above we get that the total pseudo-regret of the algorithm is bounded by

$$\mathrm{Reg}(\mathcal{A}) \leq kNr(c + B) + rk^2(c + B) + \left(1 - \frac{1}{T^4}\right)4kBT\sqrt{\frac{\log rkT}{N}} + \frac{1}{T^4}k(c + B)T.$$

Setting $N = 10\frac{T^{2/3}(\log rkT)^{1/3}}{r^{2/3}}$ we get that

$$\mathrm{Reg}(\mathcal{A}) \leq O(kr^{1/3}(c + B)(\log rkT)^{1/3}T^{2/3}).$$

$\square$

## B.1  General case

The algorithm for the case of $m = 2$ naturally extends to arbitrary correlation set sizes. Overall the structure of the algorithm remains the same where we pick a cover of $\mathcal{C}$ and estimate the losses incurred in states that belong to the cover. Using the estimated losses we are able to approximately estimate the loss of any vertex at any other state. In order to do this we extend the definition of the cover as follows. Given correlation sets of arbitrary size in $\mathcal{C}$, a vertex $v_i$ may participate in many of them. We say that vertices $v_i$ and $v_j$ share a correlation set, if they appear together in a set in $\mathcal{C}$. Consider the set of indices of all the vertices that $v_i$ shares a correlation set with. We partition this set into disjoint subsets such that no two vertices in different subsets share a correlation set. For a

given vertex $v_i$, we denote this collection of disjoint subsets by $I_i$. For example, if $\mathcal{C}$ contains sets $\{1,2\}$, $\{1,3\}$, and $\{1,4\}$, then, $I_1$ consists of the set $\{2,3,4\}$. On the other hand if $\mathcal{C}$ contains sets $\{1,2,3\}$, $\{1,3,4\}$, and $\{1,6,7\}$ then, $I_1$ consists of sets $\{2,3,4\}$ and $\{6,7\}$. For a given state $s$ and $J \in I_i$ we denote by $s(J)$ the vector $s$ restricted to indices in $J$. Notice that, in the worst case, $I_i$ will consist of a single set of size at most $\min(k-1, nm)$. However, for more structured cases (e.g, $m = 2$) we expect $I_i$ to consist of subsets of small sizes.

Given $i \in [k]$, $J \in I_i$, $b \in \{0,1\}$ and vectors $u_1, u_2$, we say that $(i, b, J, u_1, u_2)$ is a dichotomy, if there exist two states $s, s' \in \mathcal{S}$ such that: (1) $s(J) = u_1, s'(J) = u_2$, (2) $s(i) = b = s'(i)$, and (3) $s, s'$ agree in all other criteria. We call such a pair of states $s, s'$ an $(i, b, J, u_1, u_2)$ pair. We next extend the definition of a cover as follows. A subset $\mathcal{K} \subseteq S$ is called a cover of $\mathcal{C}$ if for any valid dichotomy $(i, b, J, u_1, u_2)$, there exists an $(i, b, J, u_1, u_2)$ pair $s, s' \in \mathcal{K}$. In general, we will always have a cover of size at most $n2^{mn}$. Similar to (**??**), for a valid dichotomy $(i, b, J, u_1, u_2)$, we define $X_{b,J}^{i, u_1, u_2}$ as

$$X_{b,J}^{i, u_1, u_2} := \mu_i^s - \mu_i^{s'}, \tag{7}$$

where $s, s' \in \mathcal{K}$ is an $(i, b, J, u_1, u_2)$ pair. Given the loss values in the states present in $\mathcal{K}$, we can estimate the loss of any other state using Theorem 6 stated below.

**Theorem 6.** *Let $\mathcal{K}$ be a cover for $\mathcal{C}$. Then, for any state $s \in \mathcal{S}$ and any $i \in [k]$ with $s(i) = b$, we have:*

$$\mu_i^s = \mu_i^{s''} + \sum_{J \in I_i} X_{b,J}^{i, s(J), s''(J)} \tag{8}$$

*Here $s''$ is any state in $\mathcal{K}$ with $s''(i) = b$.*

*Proof.* Let $s, s' \in \mathcal{K}$ be an $(i, b, J, u_1, u_2)$ pair. When we move from state $s$ to $s'$, the only difference between the expected losses incurred by vertex $v_i$ comes from the configuration of the vertices in $J$. Hence there at at most $2^{|J|+1}$ distinct parameters governing the expected loss incurred by vertex $i$ in a given state $s$ due to the configuration of the vertices in $J$. Denoting these parameters by $\gamma_{i,J}^{b, s(J)}$ we have

$$\mu_i^{s'} - \mu_i^s = \gamma_{i,J}^{b, s'(J)} - \gamma_{i,J}^{b, s(J)} := X_{b,J}^{i, s'(J), s(J)}.$$

Given the loss values for the states in the cover $\mathcal{K}$, we can estimate the quantities $X_{b,J}^{i, s(J), s''(J)}$.

Next, for an arbitrary state $s$ such that $s(i) = b$, let $s'' \in \mathcal{K}$ be such that $s''(i) = b$. We have

$$\mu_i^s = \mu_i^{s''} + \sum_{J \in I_i} \gamma_{i,J}^{b, s(J)} - \gamma_{i,J}^{b, s''(J)}$$
$$= \sum_{J \in I_i} X_{b,J}^{i, s(J), s''(J)}.$$

$\square$

For general correlation sets with each vertex participating in at most $n$ sets, we use (8) instead of (5) to estimate losses in step 4 of the algorithm in Figure 3. The algorithm for general $m$ is described in Figure 4 and has the following associated regret guarantee. The proof is identical to the proof of Theorem 5.

**Theorem 7.** *Consider an MDP$(\mathcal{S}, \mathcal{A}, \mathcal{C}, \boldsymbol{\theta})$ with losses bounded in $[0, B]$ and maximum cost of fixing a vertex being $c$. Given correlations sets $\mathcal{C}$ of size at most $m$, and a cover $\mathcal{K}$ of $\mathcal{C}$ of size $r \leq n2^{mn}$, the algorithm in Figure 4 achieves a pseudo-regret bounded by $O(kr^{1/3}(c + B)(\log rkT)^{1/3}T^{2/3})$. Furthermore, given access to the optimization oracle for Eq. (3) the algorithm runs in time polynomial in $k$, $n = |\mathcal{C}|$ and $r = |\mathcal{K}|$.*

Our algorithms are also scalable. During step 1 they only explore the states in the cover $\mathcal{K}$ that could be much smaller than the full state space.

> **Input:** The graph $\mathcal{G}$, correlation sets $\mathcal{C}$, fixing costs $c_i$.
> 1. Pick a cover $\mathcal{K} = \{s_1, s_2, \ldots, s_r\}$ of $\mathcal{C}$.
> 2. Let $N = 10 \frac{T^{2/3}(\log rkT)^{1/3}}{r^{2/3}}$.
> 3. For each state $s \in \mathcal{K}$ do:
>    - Move from current state to $s$ in at most $k$ time steps.
>    - Play action $a = 0$ in state $s$ for the next $N$ time steps to obtain an estimate $\widehat{\mu}_i^s$ for all $i \in [k]$.
> 4. Using the estimated losses for the states in $\mathcal{K}$ and Equation (8), run the oracle for the optimization (3) to obtain an approximately optimal state $\hat{s}$.
> 5. Move from current state to $\hat{s}$ and play action $a = 0$ from $\hat{s}$ for the remaining time steps.

Figure 4: Online algorithm for general $m$ achieving $\tilde{O}(T^{2/3})$ pseudo-regret.

## B.2 Beyond $T^{\frac{2}{3}}$ regret

In this section, we present algorithms for our problem that achieve $\tilde{O}(\sqrt{T})$ regret, first in the case $m = 1$, next for any $m$, under the natural assumption that each criterion does not participate in too many correlations sets.

Let us first point out that our problem can be cast as an instance of the stochastic multi-armed bandit problem with switching costs, where each state $s$ is viewed as an arm and where the cost of transitions from state $s$ to state $s'$ is the switching cost between $s$ and $s'$. For the instance of this problem with identical switching costs, Cesa-Bianchi et al. [2013][Appendix A] gave an algorithm achieving expected regret $\tilde{O}(\sqrt{T})$, via an arm-elimination technique with at most $O(\log \log T)$ switches. However, naturally, the regret guarantee and the time complexity of that algorithm depend on the number of arms, which in our case is exponential ($2^k$). We will show here that, in most realistic instances of our model, we can achieve $\tilde{O}(\sqrt{T})$ regret efficiently.

We first consider the case where the correlations sets in $\mathcal{C}$ are of size one ($m = 1$). In this case, the parameter vector $\boldsymbol{\theta}$ can be described using the following $2k$ parameters: for each $i \in [k]$, let $\gamma_i^0$ denote the expected loss incurred by criterion $i$ when it is unfixed and $\gamma_i^1$ its expected loss when it is fixed. In this case, the cover $\mathcal{K}$ is of size $k + 1$ and includes the all-zero state, as well as $k$ states corresponding to the indicator vectors of the $k$ vertices. Our algorithm is similar to the UCB algorithm for multi-armed bandits Auer et al. [2002] and maintains optimistic estimates of the parameters. For every vertex $i$, we denote by $\tau_{i,t}^0$ the total number of time steps up to $t$ (including $t$) during which the vertex $v_i$ is in an unfixed position and by $\tau_{i,t}^1$ the total number of times steps up to $t$ during which vertex $v_i$ is in a fixed position. Fix $\delta \in (0, 1)$ and let $\hat{\gamma}_{i,t}^b$ be the empirical expected loss observed when vertex $v_i$ is in state $b$, for $b \in \{0, 1\}$. Our algorithm maintains the following optimistic estimates at each time step $t$,

$$\tilde{\gamma}_{i,t}^b = \hat{\gamma}_{i,t}^b - 10B\sqrt{\frac{\log(kT/\delta)}{\tau_{i,t}^b}}. \tag{9}$$

To minimize the fixing cost incurred when transitioning from one state to another, our algorithm works in episodes. In each episode $h$, the algorithm first uses the current optimistic estimates to query the optimization oracle and determine the current best state $s$. Next, it remains at state $s$ for $t(h)$ time steps before querying the oracle again. The number of time steps $t(h)$ will be chosen carefully to avoid incurring the fixing costs too often. The algorithm is described in Figure 5. We will prove that it benefits from the following regret guarantee.

**Theorem 2.** *Consider an MDP$(\mathcal{S}, \mathcal{A}, \mathcal{C}, \boldsymbol{\theta})$ with losses bounded in $[0, B]$ and maximum cost of fixing a vertex being $c$. Given correlations sets $\mathcal{C}$ of size one, the algorithm of Figure 5 achieves a pseudo-regret bounded by $O(k^2(c + B)^2\sqrt{T}\log T)$. Furthermore, given access to an oracle for (3), the algorithm runs in time polynomial in $k$.*

**Input:** graph $\mathcal{G}$, correlation sets $\mathcal{C}$, fixing costs $c_i$.

1. Let $\mathcal{K}$ be the cover of size $k+1$ that includes the all zeros state and the states corresponding to indicator vectors of the $k$ vertices.
2. Move to each state in the cover once and update the optimistic estimates according to (9).
3. For episodes $h = 1, 2, \dots$ do:
   - Run the optimization oracle for solving Eq. (3) with the optimistic estimates as in (9) to get a state $s$.
   - Move from current state to state $s$. Stay in state $s$ for $t(h)$ time steps and update the corresponding estimates using (9). Here $t(h) = \min_i \tau_{i,t_h}^{s(i)}$ and $t_h$ is the total number of time steps before episode $h$ starts.

Figure 5: Online algorithm for $m = 1$ with $\tilde{O}(\sqrt{T})$ regret.

*Proof.* We first bound the total number of different states visited by the algorithm. Initially the algorithm visits $k+1$ states in the cover. After that, each time the optimization oracle returns a new state $s$, by the definition of $t(h)$, the number of time steps where some vertex is in a $0$ or $1$ position is doubled. Hence, at most $O(k \log T)$ calls are made to the optimization oracle. Noticing that one can move from one state to another in at most $k$ time steps, the total loss incurred during the switching of the states is bounded by $O(k^2(c + B) \log T)$.

For $\epsilon > 0$ to be chosen later, we consider the episodes where the algorithm plays a state $s$ with expected loss at most $\epsilon$ more than that of the best state $s^*$. The total expected regret accumulated in these *good* episodes is at most $\epsilon T$. We next bound the expected regret accumulated during the bad episodes.

From Hoeffding's inequality we have that for any time $t$, with probability at least $1 - \frac{\delta}{T^3}$, for all $i \in [k], b \in \{0, 1\}$,

$$\tilde{\gamma}_{i,t}^b + 20B \sqrt{\frac{\log(kT/\delta)}{\tau_{i,t}^b}} \geq \gamma_i^b \geq \tilde{\gamma}_{i,t}^b. \tag{10}$$

Let $G$ be the good event that (10) holds for all $t \in [1, T]$. Conditioned on $G$ we also have that for any state $s$ and vertex $i$

$$\mu_i^s \geq \tilde{\mu}_i^s, \tag{11}$$

where $\tilde{\mu}_i^s$ is the estimated loss using the optimistic estimates. We will bound the expected regret accumulated in the bad episodes conditioned on the event $G$ above.

In order to do this we define certain key quantities. Consider a particular trajectory $\mathcal{T}$ of $T$ time steps executed by the algorithm. Furthermore, let $\mathcal{T}$ be such that the good event in (10) holds during the $T$ time steps. We associate the following random variables with the trajectory. Let $N_\epsilon$ be the total number of time steps spent in bad episodes. Furthermore, let $\text{Reg}_\epsilon$ be the total accumulated regret during these time steps. Then it is easy to see that $\mathbb{E}[\text{Reg}_\epsilon | G] > \epsilon N_\epsilon$. For each vertex $v_i$ and $b \in \{0, 1\}$ we define $\tau_\epsilon(i, b)$ to be the total number of time steps that vertex $v_i$ spends in bad episodes in position $b$ and $\tau_\epsilon(i, b, t)$ to be the total number of time steps spent in bad episodes up to time step $t$. Notice that

$$\sum_b \sum_i \tau_\epsilon(i, b) \leq 2k N_\epsilon. \tag{12}$$

Consider a particular bad episode $h$ and let $s$ be the state returned by the optimization oracle during that episode. Then conditioned on the good event $G$, the total regret $\text{Reg}_h$ accumulated during episode

$h$ satisfies

$$\mathbb{E}[\mathsf{Reg}_h|\mathcal{T}] = \sum_i \left(\mu_i^s - \mu_i^{s^*}\right)t(h)$$

$$\leq \sum_i \left(\mu_i^s - \tilde{\mu}_i^{s^*}\right)t(h) \qquad \big(\text{from}(11)\big)$$

$$\leq \sum_i \left(\mu_i^s - \tilde{\mu}_i^{s}\right)t(h) \qquad \big(\text{since } s \text{ is best state according to the optimistic losses}\big)$$

$$\leq \sum_i \left(\gamma_i^{s(i)} - \tilde{\gamma}_{i,t_h}^{s(i)}\right)t(h)$$

$$\leq \sum_i 20B\sqrt{\frac{\log(kT/\delta)}{\tau_{i,t_h}^b}}t(h). \qquad \big(\text{from } (9)\big)$$

In the above inequality, the expectation is taken over the loss distribution for each vertex during states visited in the trajectory $\mathcal{T}$.

Since $\tau_{i,t_h}^b \geq \tau_\epsilon(i, b, t_h)$ we have we have that

$$\mathbb{E}[\mathsf{Reg}_h|\mathcal{T}] \leq \sum_i 20B\sqrt{\frac{\log(kT/\delta)}{\tau_\epsilon(i, b, t_h)}}t(h).$$

Summing over bad episodes, the total expected regret in bad episodes can be bounded by

$$\mathbb{E}[\mathsf{Reg}_\epsilon|\mathcal{T}] \leq \sum_i \sum_b \sum_{h:h \text{ is bad}} 20B\sqrt{\frac{\log(kT/\delta)}{\tau_\epsilon(i, b, t_h)}}t(h). \tag{13}$$

Notice that $\tau_\epsilon(i, b, t_h) = \sum_{h' < h:h' \text{ is bad}} t(h')$. Furthermore, we know that (Jaksch et al. [2010]) for any sequence $z_1, z_2, \ldots, z_h$ of non-negative numbers such that $z_i \geq 1$,

$$\sum_{i=1}^h \frac{z_i}{\sqrt{\sum_{j=1}^{i-1} z_j}} \leq (1 + \sqrt{2})\sqrt{\sum_{i=1}^h z_i}. \tag{14}$$

From (14) we get:

$$\sum_{h:h \text{ is bad}} \frac{t(h)}{\sqrt{\tau_\epsilon(i, b, t_h)}} \leq \sqrt{\tau_\epsilon(i, b)}.$$

Substituting into (13) we get that

$$\mathbb{E}[\mathsf{Reg}_\epsilon|\mathcal{T}] \leq \sum_i \sum_b 20B\sqrt{\log(kT/\delta)}\sqrt{\tau_\epsilon(i, b)}.$$

Using (12) we have that the above expected regret is maximized when $\tau_\epsilon(i, b)$ are equal, thereby implying

$$\mathbb{E}[\mathsf{Reg}_\epsilon|\mathcal{T}] \leq 20Bk\sqrt{\log(kT/\delta)}\sqrt{N_\epsilon}.$$

Using the fact that $\mathbb{E}[\mathsf{Reg}_\epsilon|G] > \epsilon N_\epsilon$ we get that conditioned on $G$,

$$N_\epsilon \leq \frac{400B^2k^2\log(kT/\delta)}{\epsilon^2}.$$

Combining trajectories $\mathcal{T}$ where the good event $G$ holds, we get that the total expected regret accumulated in the bad episodes satisfies

$$\mathbb{E}[\mathsf{Reg}_\epsilon|G] \leq 20Bk\sqrt{\log(kT/\delta)}\sqrt{N_\epsilon}$$

$$\leq 400B^2k^2\frac{\log(kT/\delta)}{\epsilon}.$$

Combining the above with the total expected regret accumulated in the good episodes, the loss of moving to different states, and the probability of good event $G$ not holding, we get

$$\mathsf{Reg}(\mathcal{A}) \leq 400B^2k^2\frac{\log(kT/\delta)}{\epsilon} + \epsilon T + \frac{k(c+B)}{T^3} + O(k^2(c+B)\log T).$$

Setting $\epsilon = \frac{1}{\sqrt{T}}$ and $\delta = \frac{1}{T^4}$, we have the final bound

$$\mathsf{Reg}(\mathcal{A}) \leq O\big((c+B)^2k^2\sqrt{T}\log(T)\big).$$

$\square$

The above result extends to higher $m$ values, assuming that each vertex does not participate in too many correlation sets. If a vertex $v_i$ appears in at most $O(\log k)$ correlation sets, then the total loss incurred by vertex $v_i$ in any state depends on the position of $v_i$ and every other vertex that it is correlated with. Hence the total loss incurred by vertex $v_i$ depends on an $O(m \log k)$-dimensional vector. For every configuration $\boldsymbol{b}$ of this vector, we associate with each vertex $v_i$, parameters $\gamma_i^{\boldsymbol{b}}$. Notice that there are at most $O(k^m)$ such parameters. Each parameter is in turn a sum of a subset of the parameters in $\boldsymbol{\theta}$. Notice that in this case the size of the cover $\mathcal{K}$ is upper bounded by $O(k^{m+1})$. Our algorithm for higher $m$ values is similar to the one for $m = 1$, but instead maintains optimistic estimates of the parameters $\gamma_i^{\boldsymbol{b}}$ via

$$\tilde{\gamma}_{i,t}^{\boldsymbol{b}} = \hat{\gamma}_{i,t}^{\boldsymbol{b}} - 10B\sqrt{m\frac{\log(kT/\delta)}{\tau_{i,t}^{\boldsymbol{b}}}}. \tag{15}$$

Here $\tau_{i,t}^{\boldsymbol{b}}$ is the total time spent up to and including $t$ where the vertex $i$ and the vertices that it is correlated with are in configuration $\boldsymbol{b}$. Similarly, for a given state $s$, we will denote by $\boldsymbol{s}(i)$, the configuration of the vertex $i$ and the vertices that it is correlated with. The algorithm is sketched below

---

**Input:** The graph $\mathcal{G}$, correlation sets $\mathcal{C}$, fixing costs $c_i$.

1. Let $\mathcal{K}$ be the cover of size $O(k^{m+1})$.
2. Move to each state in the cover once and update the optimistic estimates according to (15).
3. For episodes $h = 1, 2, \ldots$ do:
    - Run the optimization oracle (3) with the optimistic estimates as in (15) to get a state $s$.
    - Move from current state to state $s$. Stay in state $s$ for $t(h)$ time steps and update the corresponding estimates using (15). Here $t(h) = \min_i \tau_{i,t_h}^{\boldsymbol{s}(i)}$ and $t_h$ is the total number of time steps before episode $h$ starts.

---

Figure 6: Online algorithm for higher $m$.

For $m \geq 1$, we obtain the following guarantee.

**Theorem 8.** *Consider an MDP$(\mathcal{S}, \mathcal{A}, \mathcal{C}, \boldsymbol{\theta})$ with losses bounded in $[0, B]$ and maximum cost of fixing a vertex being $c$. Given correlations sets $\mathcal{C}$ of size at most $m$ such that each vertex participates in at most $O(\log k)$ sets, the the algorithm in Figure 6 achieves a pseudo-regret bounded by $O(mk^{2m+2}(c+B)^2\sqrt{T}\log T)$. Furthermore, given access to an oracle for (3), the algorithm runs in time polynomial in $O(k^{m+1})$.*

*Proof.* The proof is very similar to the proof of Theorem 2. Since each time the optimization oracle is called the time spent in some configuration $\boldsymbol{s}(i)$ is doubled, we get that the total number of calls to the optimization oracle are bounded by $O(k^m \log T)$. Hence the total loss incurred during the exploration phase can be bounded by $O(k^m(c+B)\log T)$. Let $G$ be the good event that (15) holds for all $t \in [1, T]$.

As before, the loss incurred during good episodes is bounded by $\epsilon T$. Define $\tau_\epsilon(i, \boldsymbol{b})$ to be the total time that vertex $i$ and vertices that it is correlated with spend in configuration $\boldsymbol{b}$ during bad episodes. Then analogous to (12) we have

$$\sum_{\boldsymbol{b}} \sum_i \tau_\epsilon(i, \boldsymbol{b}) \leq O(k^m)N_\epsilon.$$

For a trajectory $\mathcal{T}$ where the good event $G$ holds, the total expected regret in bad episodes can be bounded as

$$\mathbb{E}[\text{Reg}_\epsilon | \mathcal{T}] \leq \sum_i \sum_{\boldsymbol{b}} \sum_{h: h \text{ is bad}} 20 B \sqrt{m \frac{\log(kT/\delta)}{\tau_\epsilon(i, \boldsymbol{b}, t_h)} t(h)} \tag{16}$$

$$\leq \sum_i \sum_{\boldsymbol{b}} 20 B \sqrt{m \log(kT/\delta)} \sqrt{\tau_\epsilon(i, \boldsymbol{b})} \tag{17}$$

$$\leq O(Bk^{m+1}) \sqrt{m \log(kT/\delta)} \sqrt{N_\epsilon}. \tag{18}$$

Using the fact that $\mathbb{E}[\text{Reg}_\epsilon | \mathcal{T}] > \epsilon N_\epsilon$ we get that for a trajectory where the event $G$ holds,

$$N_\epsilon \leq \frac{O(R^2 k^{2m+2} m \log(kT/\delta))}{\epsilon^2}.$$

Hence we get that conditioned on the good event $G$, the total expected regret accumulated in the bad episodes is at most

$$\mathbb{E}[\text{Reg}_\epsilon | G] \leq O\left(R^2 m k^{2m+2} \frac{\log(kT/\delta)}{\epsilon}\right).$$

Combining the above with the total expected regret accumulated in the good episodes, the loss of moving to different states, and the probability of the event $G$ not holding we get

$$\text{Reg}(\mathcal{A}) \leq O\left(B^2 m k^{2m+2} \frac{\log(kT/\delta)}{\epsilon}\right) + \epsilon T + \frac{k(c + B)}{T^3} + O(k^m \log T).$$

Setting $\epsilon = \frac{1}{\sqrt{T}}$ and $\delta = \frac{1}{T^4}$, we have the final bound

$$\text{Reg}(\mathcal{A}) \leq O\left((c + B)^2 m k^{2m+2} \sqrt{T} \log(T)\right).$$

$\square$

An important corollary of the above is the following

**Corollary 1.** *If $\mathcal{G}$ is a constant degree graph with correlation sets consisting of subsets of edges in $\mathcal{G}$, then there is a polynomial time algorithm that achieves a pseudo-regret bounded by $O(k^6(c + B)^2 \sqrt{T} \log T)$.*

**Modeling assumptions and extensions**. Here we briefly discuss assumptions and extensions.

**Scalability**. The running time of our algorithms depends linearly on the size of the cover $\mathcal{K}$. While in the worst case the size of the cover could be exponential in $n, m$, in practice, we expect it to be rather small, in which case our algorithms are quite efficient.

**Loss function**. The choice of the loss function is critical in the effectiveness of our model. We made the simplifying assumption that the loss at each time step is additive in the losses incurred by correlation sets. A careless choice of what the additive losses correspond to may result in a sub-optimal overall. For example, a poor choice is one that uses the volume of complaints, i.e., how many complaints have triggered a criterion. This will make us vulnerable to the loudest voices in the system. In Section D, we discuss how our framework can be implemented in practice and present reasonable choices for the loss function. We further discuss the choice of the loss function in the case of the COMPAS example in Appendix E.

**Adversarial manipulation.** Our model may be vulnerable to strategic coordination. A malicious group of users can form a sub-community generating a large number of complaints to press the system to include a new criterion in the graph. The presence of such poor criteria may result in an overall suboptimal system. Modeling this scenario is beyond the scope of the current work.

**Continuous states**. This is a direction for future work.

## B.3 Additional proofs for the Stochastic setting

Here we show that in the stochastic model, if correlation sets are of size one then one can efficiently approximate the cost of the optimal state up to a factor of two.

**Theorem 9.** *If correlations sets are of size one ($m = 1$), then, for any $\epsilon, \delta > 0$, the true parameter vector for MDP$(\mathcal{S}, \mathcal{A}, \mathcal{C}, \boldsymbol{\theta})$ can be approximated to $\epsilon$-accuracy in $\ell_\infty$-norm with probability at least $1 - \delta$, in at most $O(\frac{B^2 k}{\epsilon^2} \log(\frac{k}{\delta}))$ time steps and exploring at most $k + 1$ specific states in $\mathcal{S}$. Furthermore, given a parameter vector $\boldsymbol{\theta}$, there is an algorithm that runs in time polynomial in $k$ and finds an approximately optimal state $s'$ such that $g(s') \le 2 \min_{s \in \mathcal{S}} g(s)$.*

*Proof.* Notice that when correlation sets are of size one, the expected loss incurred for criterion $v_i$ at any given state $s$ solely depends on whether $s(i) = 0$ or $s(i) = 1$. Hence in this case the MDP consists of $2k$ parameters where we use $\gamma_i^1$ and $\gamma_i^0$ to denote the expected losses incurred by vertex $i$ when it is in fixed and unfixed position respectively. For any $\delta > 0$, by Hoeffding's inequality, we have that if we stay in state $s = (0, 0, \ldots, 0)$ for $N = \frac{B^2}{\epsilon^2} \log(2k/\delta)$ time steps then with probability at least $1 - \frac{\delta}{2}$, we have each $\gamma_i^0$ estimated up to $\epsilon$ accuracy. Let $e_i \in \{0, 1\}^k$ denote the indicator vector for $i$. If we stay in state $s = e_i$ for $\frac{B^2}{\epsilon^2} \log(2k/\delta)$ time steps, then with probability at least $1 - \frac{\delta}{2}$ we have $\gamma_i^1$ estimated up to $\epsilon$ accuracy. Hence, overall after $O(\frac{B^2 k}{\epsilon^2} \log(\frac{k}{\delta}))$ time steps, we have each parameter estimated up to $\epsilon$ accuracy. Notice that in total we observe at most $k + 1$ states.

Next we show how to efficiently approximate the loss of the best state. Given the parameters of the MDP each vertex has two costs $\Lambda_i^{(1)} = \gamma_i^0$, denoting the cost incurred if the vertex is unfixed and $\Lambda_i^{(2)} = c_i + \gamma_i^1$, denoting the cost incurred if the vertex is fixed. Without loss of generality assume that $\Lambda_i^{(1)} > \Lambda_i^{(2)}$ (any vertex that does not satisfy this can be safely left unfixed). For each $i$, define $y_i = 1$ if vertex $i$ is unfixed otherwise define $y_i = 0$. Then the offline problem of finding the best state can be written as

$$\min \sum_{i=1}^{k} (1 - y_i) \Lambda_i^2 + y_i \Lambda_i^1 = \sum_{i=1}^{k} y_i \gamma_i + \sum_{i=1}^{k} \Lambda_i^{(2)}$$
$$\text{s.t. } y_i \in \{0, 1\}$$
$$y_i + y_j \ge 1, \ \forall (v_i, v_j) \in E.$$

Here $\gamma_i = \Lambda_i^{(1)} - \Lambda_i^{(2)} > 0$. By relaxing $y_i$ to be in $[0, 1]$ and solving the corresponding linear programming relaxation, we get a solution $y_1^*, y_2^*, \ldots, y_k^*$. Let LPval denote the linear programming objective value achieved by $y_1^*, y_2^*, \ldots, y_k^*$. Since the linear programming formulation is a valid relaxation of the problem of finding the best state, we have LPval $\le \min_{s \in \mathcal{S}} g(s)$.

We output the state $s'$ in which a vertex $i$ if and only if $y_i^* < 1/2$. Let $S$ be the set of fixed vertices. We have

$$g(s') = \sum_{i \in S} \Lambda_i^{(2)} + \sum_{i \notin S} \Lambda_i^{(1)}$$
$$= \sum_{i=1}^{k} \Lambda_i^{(2)} + \sum_{i \notin S} (\Lambda_i^{(1)} - \Lambda_i^{(2)})$$
$$= \sum_{i=1}^{k} \Lambda_i^{(2)} + \sum_{i \notin S} \gamma_i$$
$$< \sum_{i=1}^{k} \Lambda_i^{(2)} + 2 \sum_{i \notin S} y_i^* \gamma_i$$
$$< 2 \Big( \sum_{i=1}^{k} \Lambda_i^{(2)} + \sum_{i=1}^{k} y_i^* \gamma_i \Big)$$
$$< 2 \cdot \text{LPval}$$
$$\le \min_{s \in \mathcal{S}} 2 g_{\boldsymbol{p}}(s).$$

$\square$

## C  Adversarial setting

In the previous section, we studied a stochastic model for arrival of complaints and designed no regret algorithms. In this section, we study the setting when we cannot make assumptions about the arrival of complaints. In particular, we study an adversarial model where at each time step multiple complaints arrive for the vertices in $\mathcal{G}$ via the choice made by an oblivious adversary. For a given vertex $v_i$ and time step $t$, we denote by $\ell_{i(t)}$ the loss incurred if criterion $v_i$ is unfixed at time $t$. Similar to the setting from the previous section, initially all the vertices in $\mathcal{G}$ are in unfixed state and each vertex has a fixing cost of $c_i$. At each time step the algorithm can decide to fix a particular vertex. As a result all its neighbors get unfixed. At time step $t$, if criterion $v_i$ is unfixed then the the algorithm incurs a loss of $\ell_{i(t)}$. If $v_i$ is fixed at time step $t$ then algorithm incurs no loss. The overall loss incurred by the algorithm is the total fixing cost and the total loss incurred over the arrival complaints. As before, we will denote a configuration of the vertices in $\mathcal{G}$ using a vector $s \in \{0,1\}^k$ with $s(i) = 0$ representing an unfixed vertex. For an algorithm $\mathcal{A}$ processing the request sequence, During the course of $T$ time steps, the total loss of processing the complaints is

$$\text{Loss}(\mathcal{A}) = \sum_{i=1}^{k} \sum_{t=1}^{T} \ell_{i(t)} \cdot \mathbb{1}(s_t(i) = 0) + \sum_{i=1}^{k} \sum_{t=2}^{T} c_i \cdot \mathbb{1}(s_{t-1}(i) = 0, s_t(i) = 1). \tag{19}$$

Define OPT to be the algorithm that given the entire loss sequence in advance, makes the optimal choice of decisions to fix vertices. Following standard terminology we define the *competitive ratio* of an algorithm $\mathcal{A}$ to be the maximum of $\text{Loss}(\mathcal{A})/\text{Loss}(\text{OPT})$ over all possible complaint sequences. We will design efficient online algorithms for processing the complaints that achieve a constant competitive ratio. Notice that in this setting, in order for the competitive ratio to be finite, we need to bound the range of the losses and the fixing costs of the vertices. We will assume that the cost of fixing each vertex is at least one and as before assume that the losses are bounded in the range $[0, B]$. For ease of exposition, in the rest of the discussion we will assume that at each time step complaints arrive for one of the vertices in $\mathcal{G}$. A simple reduction shows that an algorithm that is competitive with OPT in this setting remains so in the general setting with the same competitive ratio. We discuss this at the end of the section. Via this reduction we can consider the loss sequence to be of the form $((i_1, \ell_{i_1}), \ldots, (i_T, \ell_{i_T}))$ where $i_t$ is the index of the criterion for which the $t$th complaint arrives and $\ell_{i_t}$ is the associated loss.

To get a better understanding of the above adversarial setting, consider the case when the graph $\mathcal{G}$ over the criteria has no edges, i.e., there are no conflicts. In this case, given a sequence of complaints, each with unit loss value, the optimal offline algorithm that has the entire loss sequence in advance can independently make a decision for each vertex. In particular, if the total loss of the complaints incurred at vertex $v_i$ exceeds the fixing cost $c_i$ then the optimal decision is to fix the vertex $v_i$, and otherwise simply incur the loss from the arriving complaints. In this case the online algorithm can also simply process each vertex independently. At each vertex the algorithm is faced with the classical *ski-rental* problem for which there exists a deterministic algorithm that is 2-competitive with optimal algorithm Karlin et al. [1988]. For each vertex $i$, the online algorithm simply waits till a total loss of $c_i$ or more has been incurred on vertex $i$ and then decides to fix it. It is easy to see that the total cost incurred by this strategy is at most twice the cost incurred by OPT.

However, the above algorithm will fail miserably in the presence of conflicts in the graph $\mathcal{G}$. As an example consider a graph with two vertices $v_i$ and $v_j$ that are connected by an edge. Let the fixing cost of $v_i$ be 1 and the fixing cost of $v_j$ be $C \gg 1$. Consider a sequence of complaints, each of unit loss, consisting of $C$ complaints for $v_j$ followed one complaint for $v_i$. If this sequence is repeated $T$ times the optimal offline algorithm OPT incurs a loss of $C + T$ by fixing $v_j$ and incurring losses due to $v_i$. However, the algorithm above will incur a cost of $(2C + 2)T$ thereby leading to an unbounded competitive ratio. Hence, in order to achieve a good competitive ratio one must make decisions not only based on the loss incurred at the given vertex $v_i$, but also the status of the vertices in the neighborhood of $v_i$. Our main result in this section is the algorithm in Figure 7 that achieves a constant factor competitive ratio.

The algorithm described in Figure 7 makes decisions based on local neighborhood information of a vertex. Intuitively, if a vertex is fixed only once or a few times in the optimal algorithm one would like to avoid fixing it too many times. In order to achieve this, each time a vertex $v_i$ is fixed, it adds a barrier of $\kappa_i = c_i$ to the loss any of its neighbors need to incur before getting fixed. Hence, if a vertex

**Input:** The graph $\mathcal{G}$, fixing costs $c_i$, loss sequence $(i_1, \ell_{i_1}), \ldots, (i_T, \ell_{i_T})$.

1. For each $i \in [k]$, initialize $\tau_i, \kappa_i$ to 0.
2. Process the complaints in sequence and for each complaint $(i, \ell_i)$ such that $v_i$ is unfixed do:
   (a) $\tau_i = \tau_i + \ell_i$.
   (b) While $\ell_i > 0$ and exists $j \in N(i)$ with $\kappa_j > 0$ do:
      i. Set $\Delta = \min(\ell_i, \kappa_i)$ and reduce both $\kappa_i$ and $\ell_i$ by $\Delta$.
   (c) If $\tau_i \geq \max\left(c_i, \sum_{j \in N(i)} \kappa_j\right)$ fix $v_i$. Set $\tau_i$ to 0 and $\kappa_i$ to $c_i$. Set $\tau_j = 0$ for all $j \in N(i)$.

Figure 7: Online algorithm for the adversarial setting.

is connected to a lot of fixed vertices then it has a high barrier to cross before getting fixed. During the course of the algorithm each unfixed vertex is in one of the two phases. In phase one, the vertex is accumulating losses to pay for the barrier introduced by its neighbors (step 2(b) of the algorithm). In phase two, once the barrier has been crossed the vertex follows the standard ski-rental strategy independent of other vertices for making a decision as to fix or not. Notice that via step 2(b) of the algorithm, multiple neighbors of a vertex $v_i$ can help bring down the barrier of $c_i$ introduced by the action of fixing vertex $v_i$. This is necessary to ensure the online algorithm does not incur a large loss on a vertex by waiting too long to fix it.

As an example consider a graph $\mathcal{G}$ with $k$ vertices and $k - 1$ edges, where vertex $v_0$ is the central vertex connected to every other vertex. Let the fixing cost of vertex $v_0$ be a large value $C$, and the fixing cost of other vertices be one. We consider a sequence of $C$ complaints, each with unit loss arriving for vertex $v_0$, followed by a sequence of $C$ complaints for vertex $v_1$ and so on. In this case the optimal offline solution incurs a loss of $C + k$ by deciding to fix every vertex except $v_0$. After processing $C$ complaints for $v_0$, the online algorithm will fix $v_0$ and incur a loss of $2C$. Next, during the course of processing $C$ complaints for $v_1$, the algorithm fixes $v_1$ and incurs an additional loss of $C + 1$. More importantly, due to step 2(b), the barrier $\kappa_0$ introduced by vertex $v_0$ has been reduced to zero and hence the algorithm only incurs a loss of 2 per vertex for the remaining sequence for a total loss of $3C + 2k - 1$. Without the presence of step 2(b) each vertex will incur a loss of $C$ leading to a large competitive ratio.

Notice that our algorithm in Figure 7 is designed for a setting where in each time step complaints arrive for a single vertex in $\mathcal{G}$. If multiple vertices accumulate complaints in a time step, we can simply order them arbitrarily and run the algorithm on the new sequence. Let OPT be the optimal offline algorithm according to the chosen ordering of the complaints. Let OPT' be the optimal offline algorithm when processing multiple complaints per time step. Notice that for each time step, the loss of OPT cannot be larger than that of OPT' since any choice available to OPT' is available to OPT as well. Hence it is enough to design an algorithm that is competitive with OPT. In particular, we have the following theorem.

**Theorem 3.** *Let $\mathcal{G}$ be a graph with fixing costs at least one. Then, the algorithm of Figure 7 achieves a competitive ratio of at most $2B + 4$ on any sequence of complaints with loss values in $[0, B]$.*

*Proof.* Recall that $\ell_{i(t)}$ denotes the loss incurred by vertex $v_i$ at time $t$. We divide this loss into the amount that was used to reduce the $\kappa_j$ value of one its neighbors and the rest. Formally, for every edge $(i, j)$ we define $\delta_{i \to j}^t$ as follows. If in time step $t$, the complaint arrived for vertex $i$ and step 2(b) was executed to reduce $\kappa_j$ by $\Delta$, then we define $\delta_{i \to j}^t = \Delta$. Otherwise we define $\delta_{i \to j}^t$ to be zero. We also define

$$\delta_{i \to i}^t = \ell_{i(t)} - \sum_{j \in N(i)} \delta_{i \to j}^t. \tag{20}$$

If vertex $v_i$ is fixed $f_i$ times during the course of the algorithm then we have that the total loss incurred by the algorithm can be written as

$$\text{Loss}(\mathcal{A}) = \sum_{i=1}^{k} f_i c_i + \sum_{i=1}^{k} \sum_{t=1}^{T} \left(\delta_{i \to i}^t + \sum_{j \in N(i)} \delta_{i \to j}^t\right). \tag{21}$$

Next we notice that each time a vertex $v_i$ is fixed it accumulates a value of $\kappa_i = c_i$. Furthermore, the total loss incurred by vertices as a result of executing step 2(b) is upper bounded by the total $\kappa$ value accumulated. Hence we have

$$\sum_{t=1}^{T} \sum_{i=1}^{k} \sum_{j \in N(i)} \delta_{i \to j}^t \leq \sum_{i=1}^{k} f_i c_i. \tag{22}$$

Substituting into (21) we have

$$\text{Loss}(\mathcal{A}) \leq \sum_{i=1}^{k} 2 f_i c_i + \sum_{i=1}^{k} \sum_{t=1}^{T} \delta_{i \to i}^t. \tag{23}$$

Next we bound the above loss for each vertex separately. For a given vertex $v_i$ that is fixed $f_i$ times by the algorithm, we can divide the time steps into $f_i + 1$ intervals consisting of an interval $I_0$ starting from $t = 0$ up to (and including) the first time $v_i$ is fixed. The next $f_i$ intervals correspond to the time spent by $v_i$ between two successive fixes. Denoting these intervals as $I_0, I_1, \ldots$ we have that

$$2 f_i c_i + \sum_{i=1}^{k} \sum_{t=1}^{T} \delta_{i \to i}^t = \sum_{t \in I_0} \delta_{i \to i}^t + \sum_{t \in I_r} (2 c_i + \delta_{i \to i}^t). \tag{24}$$

Next we compare the above to the loss incurred by OPT for vertex $v_i$. Let $\ell_{i(t)}^*$ be the loss incurred by OPT for vertex $v_i$ at time $t$. We will denote by $s_t^*$ the state of the vertices at time $t$ according to OPT.

We instead redefine the loss incurred by OPT for vertex $v_i$ at time $t$ to be

$$\tilde{\ell}_{i(t)} = \ell_{i(t)}^* + \sum_{j \in N(i)} \delta_{j \to i}^t \mathbb{1}(s_t^*(j) = 0). \tag{25}$$

Notice that

$$\sum_{i \in N(j)} \delta_{j \to i}^t \mathbb{1}(s_t^*(j) = 0) \leq \ell_{j(t)}^*.$$

Hence we get that

$$\sum_{i=1}^{k} \sum_{t=1}^{T} \tilde{\ell}_{i(t)} \leq \sum_{i=1}^{k} \Big( \sum_{t=1}^{T} \ell_{i(t)}^* + \sum_{j \in N(i)} \ell_{j(t)}^* \Big) \tag{26}$$

$$\leq 2 \cdot \text{Loss}(\text{OPT}). \tag{27}$$

Next we consider each interval in (23) separately. For any interval $I_r$ we have that

$$\sum_{t \in I_r} \delta_{i \to i}^t \leq B c_i. \tag{28}$$

This is because after incurring a loss of more than $c_i$, any additional loss incurred by $v_i$ is due to step 2(b), since otherwise step 2(c) will be executed and $v_i$ will be fixed.

Next consider interval $I_0$. The loss incurred by the algorithm on vertex $v_i$ equals $\sum_{t \in I_0} \delta_{i \to i}^t \leq B c_i$. Either OPT fixes $v_i$ at least once during this interval or incurs the total loss. Either way we have that the loss incurred by OPT is at least

$$\min \Big( c_i, \sum_{t \in I_0} \delta_{i \to i}^t \Big) \geq \frac{\sum_{t \in I_0} \delta_{i \to i}^t}{B}. \tag{29}$$

Next consider an interval $I_r$ between two successive fixes. The loss incurred by the algorithm for vertex $v_i$ during this interval is at most

$$\sum_{t \in I_r} \delta_{i \to i}^t + 2 c_i \leq (B + 2) c_i.$$

If OPT fixes $v_i$ at least once during this interval then it incurs a cost of $c_i$. If $v_i$ remains unfixed in OPT during the course of the interval then OPT incurs a loss of at least $c_i$. This is because vertex $v_i$

went from being unfixed to fixed during the second half of the interval and hence a total loss of at least $c_i$ must have arrived for the vertex $v_i$ during this interval.

Finally, suppose vertex $v_i$ is fixed in OPT before the start of the interval and remains so throughout. Since $v_i$ goes from being fixed to unfixed during the first half of the interval, we must have $\sum_{t \in I_r} \sum_{j \in N(i)} \delta_{j \to i}^t \geq c_i$. Furthermore, since $v_i$ is fixed by OPT during this interval, OPT must incur a loss on all neighbors of $j$. In particular, from (25) we have

$$\sum_{t \in I_r} \tilde{\ell}_{i(t)} \geq \sum_{t \in I_r} \sum_{j \in N(i)} \delta_{j \to i}^t \mathbb{1}(s_t^*(j) = 0) \tag{30}$$

$$\geq c_i. \tag{31}$$

In either of the three cases we have that the loss $\sum_{t \in I_r} \tilde{\ell}_{i(t)}$ incurred by OPT is at least a $1/(B+2)$ fraction of the loss incurred by the algorithm. Summing over all the vertices and the corresponding intervals, we get that the total loss incurred by the algorithm can be bounded by

$$\text{Loss}(\mathcal{A}) \leq (B+2) \sum_{t=1}^{T} \sum_{i=1}^{k} \tilde{\ell}_{i(t)} \leq 2(B+2)\text{Loss}(\text{OPT}).$$

$\square$

# D  Experiments

In this appendix we present experimental results demonstrating the practical applicability of our proposed framework and algorithms. We view our work as primarily theoretical and of course a more extensive empirical evaluation is a direction for future work. Regarding the choice of baselines, we are not aware of any efficient algorithms that directly apply to our setting. Existing general algorithms for solving MDPs will not scale to our setting since their complexity is proportional to the number of states. Note that in our experiments we will demonstrate that our proposed algorithms can compete with the offline optimal (the best solution in hindsight) which is a strong comparison point.

## D.1  Experiments with simulated data

We evaluate the performance of our algorithms developed in the stochastic setting of Section 3. We first detail experiments on simulated data. We consider a simulated environment where the conflict graph $\mathcal{G}$ is generated from the Erdős-Renyi model: $G(k, p)$ where we set $p = 2\frac{\log k}{k}$. This ensures that with high probability $\mathcal{G}$ is connected. Next we generate correlation sets $\mathcal{C}$ consisting of pairs of vertices in $\mathcal{G}$ sampled uniformly at random. For a parameter $\alpha > 0$ that we vary, we choose $\alpha k$ pairs of vertices at random and add them as correlation sets in $\mathcal{C}$. Hence on average, each vertex participates in $\alpha$ correlation sets. We also add to $\mathcal{C}$ singleton sets for each vertex in $\mathcal{G}$. The fixing cost of each vertex is samples uniformly at random in the range $[1, 5]$.

Next we describe the choice of parameters governing the loss distribution of the different states in the MDP. For a correlation set $\{i\}$ of size one corresponding to vertex $v_i$, we sample a parameter $\gamma_i^1$ from the beta distribution Beta$(0.5, 0.5)$. For a given state $s$ with $s(i) = 1$, the loss generated due to $\{i\}$ is drawn from an exponential distribution with mean $\gamma_i^1$. For a given state $s$ with $s(i) = 0$, the loss generated due to $\{i\}$ is drawn from an exponential distribution with mean $\lambda \gamma_i^1$, where $\lambda > 1$ is a parameter that we vary. For a correlation set $\{i, j\}$ of size two, we generate two parameters $\gamma_{i,j}^{1,1}$ and $\gamma_{i,j}^{1,0}$ from the beta distribution Beta$(0.5, 0.5)$ such that $\gamma_{i,j}^{1,0} > \gamma_{i,j}^{1,1}$. For a given state $s$ with $s(i) = 1$ and $s(j) = 1$, the loss generated due to $\{i, j\}$ is drawn from an exponential distribution with mean $\gamma_{i,j}^{1,1}$. For states where $s(i) = 0$ and $s(j) = 1$ or vice-versa, the loss is generated from an exponential distribution with mean $\gamma_{i,j}^{1,0}$. Finally, for states where both $s(i) = 0$ and $s(j) = 0$, the loss is generated from an exponential distribution with mean $\lambda \gamma_{i,j}^{1,0}$.

In general, computation of the optimal state in (??) requires time exponential in $k$. In our experiments we approximate the optimal state by a linear programming relaxation of the optimization in (??) and use the appropriately rounded linear programming relaxation solution as a proxy for the optimal state.

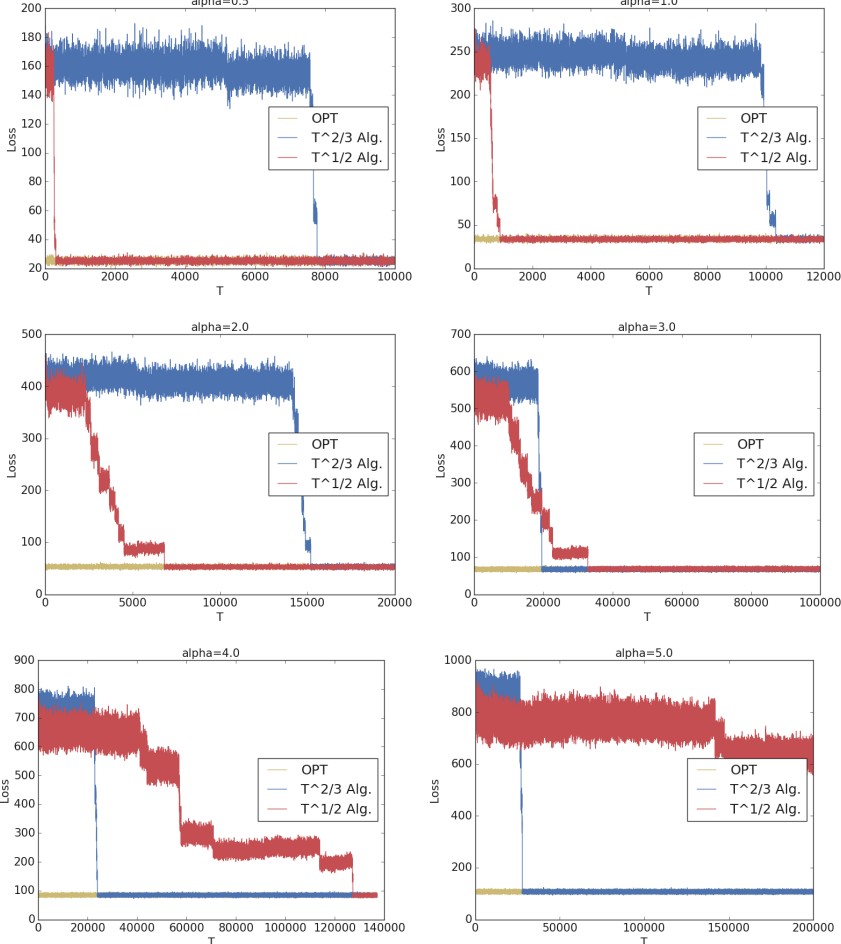

Figure 8: The figure shows the total accumulated loss incurred by the Algorithms in Figure 3 and Figure 6 on a graph with $k = 50$ criteria. The parameter $\alpha$ controls the total number of correlation sets. For each value of $\alpha$, we add $\alpha k$ random pairs of vertices into correlation sets.

For general $m$, our proposed algorithms in Figure 3 and Figure 6 have complementary strengths. While the algorithm in Figure 3 incurs a higher regret as a function of the number of time steps $T$, its running time has a polynomial dependence on the parameter $\alpha$, i.e., the number of correlation sets that a vertex participates in, on average. The algorithm in Figure 6 incurs a smaller regret of $\tilde{O}(\sqrt{T})$ as a function of $T$ at the expense of an exponential dependence on $\alpha$. In Figures 8 and 9 we empirically demonstrate this behavior where for small values of $\alpha$, the $\tilde{O}(\sqrt{T})$-regret algorithm is much better, whereas for higher values of $\alpha$ the $\tilde{O}(T^{2/3})$-regret algorithm is more desirable.

For the case of $m = 1$ however, i.e., singleton correlation sets, the algorithm in Figure 6 achieves a smaller regret and runs in polynomial time and hence is expected to outperform the explore-exploit based algorithm from Figure 3. As can be seen from Figure 10 this is indeed the case and the $\tilde{O}(\sqrt{T})$ regret algorithm significantly outperforms the $\tilde{O}(T^{2/3})$ regret algorithm.

## D.2 Experiments with a real-world dataset

In this section we demonstrate via experiments how our framework and algorithms can be applied to real world data. In order to do this we study the UCI Adult dataset [Kohavi, 1996]. The dataset comprises of $48852$ examples each represented using $124$ features, after binarizing categorical features. Each data point corresponds to a person and the label is a $0/1$ value representing whether the income of the person is more or less than \$50,000. The dataset contains information about sensitive

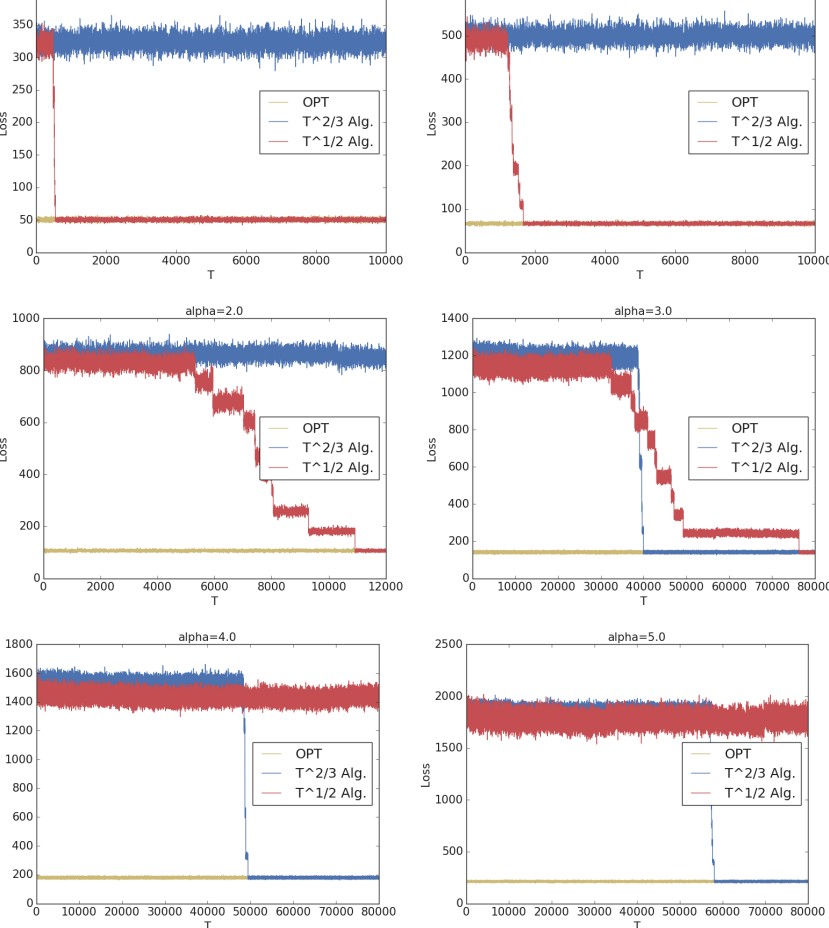

Figure 9: The figure shows the total accumulated loss incurred by the Algorithms in Figure 3 and Figure 6 on a graph with $k = 100$ criteria. The parameter $\alpha$ controls the total number of correlation sets. For each value of $\alpha$, we add $\alpha k$ random pairs of vertices into correlation sets.

attributes such as race and gender. We will simulate an online scenario where a classifier is making predictions on the income of individuals. At each time step a batch of complaints arrive, the system incurs a loss and responds by transitioning to a different state (and updating the classifier). We next describe how we instantiate various components of our stochastic model from Section 3.

*Graph $\mathcal{G}$:* We take race as a sensitive attribute that takes values in {black, white}, to obtain two sub-populations and consider two natural criteria namely the true positive rate and the AUC score. This leads to four vertices $tpr_w, tpr_b, auc_w, auc_b$. Furthermore, we add the classifier accuracy as another criterion. This leads to total 5 vertices in the graph.

*Losses and Correlation Sets:* We consider correlation sets of size one, and hence the total loss incurred at any state is the sum of the losses incurred by each criterion. For the accuracy criterion we simply define the loss to be the error of the system (the classifier). We next describe how we define the loss for the $tpr_w$ criterion. We first compute the overall true positive rate of the classifier and also the true positive rate on the white population. If the two deviate by more than a threshold $\tau$, then we penalize the classifier linearly in the violation. Therefore the loss for $tpr_w$ is defined as: $\max(0, |tpr_{overall} - tpr_w| - \tau)$. The loss for all other criteria is defined the same way. In our experiments we choose $\tau = 0.005$. Note that while we fix the threshold apriori, our method does indeed offer a way to choose the thresholds themselves in a data-driven manner. This can be achieved by simply adding, for each metric $i$, additional metrics to the graph with different thresholds $\tau_{i,1}, \tau_{i,2}, \ldots$ and so on.

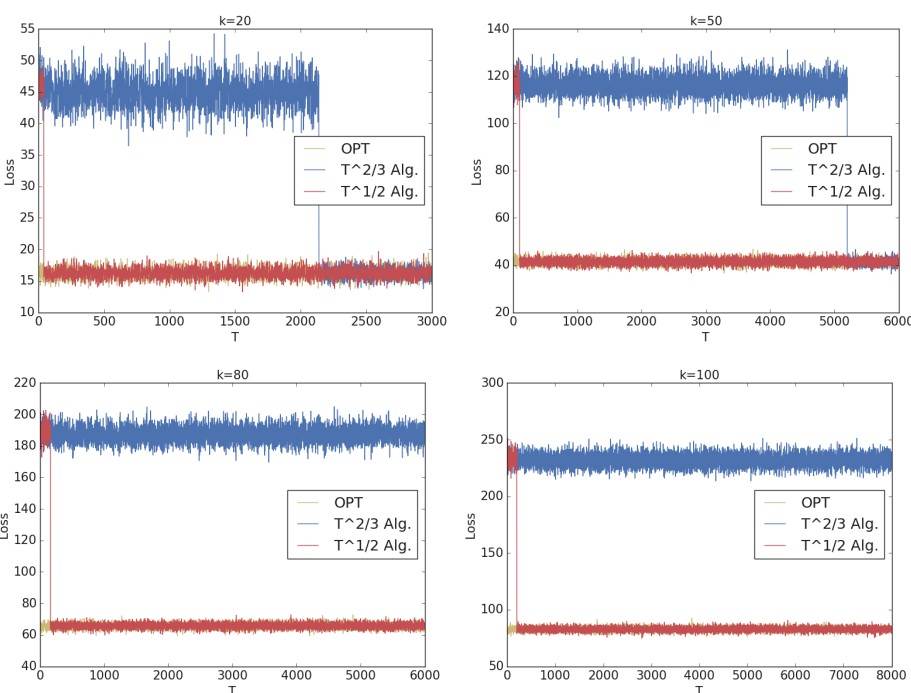

Figure 10: The figure shows the total accumulated loss incurred by the Algorithms in Figure 3 and Figure **??** for the case of $m = 1$ and varying graph sizes.

*Incompatibilities and State Transitions:* To generate incompatibilities among criteria we compute a set of valid and invalid states as follows. For each state $s \in \{0,1\}^5$, we solve a constrained optimization problem on a training set to compute a classifier. We then evaluate the classifier on the test set to compute the loss of each criterion. If the loss of any criterion is more than a specific threshold then we consider the state as an invalid state, otherwise the state is valid. In our experiments we set a threshold of $0.4$ for the accuracy criterion. For the considered criteria we present results for two thresholds, $2\tau$ and $6\tau$, the first one resulting in 4 valid states and other second one resulting in 7 valid states. To solve a constrained optimization problem we use the tensorflow constrained optimization toolkit [Cotter et al., 2018a,b]. We use the default parameter settings provided by the toolkit. The toolkit is released under Apache license 2.0. If a state $s$ has accuracy criterion set to 1, then we optimize for model accuracy subject to constraints for the other criteria that are set to 1 in $s$. If the accuracy criterion is set to 0 then we optimize for a constant loss function subject to constraints. Recall that our proposed algorithms function by fixing a criterion and as a result moving to another state. We obtain these state transitions as follows. If the algorithm asks to fix criterion $v_i$ in state $s$, we set $s(i) = 1$ to go to the next state $s'$. While $s'$ is invalid, we unfix the criterion (not including $v_i$) with the highest loss in the state $s'$ to reach another state.

*Fixing Cost:* We simply take the fixing cost of each criterion to be 1.

*Simulating Complaints:* We divide the dataset into a set of 16000 examples that we use to update our classifier at each time step and the remaining *test* set to simulate the arrival of complaints. At each time step, we randomly select a batch of examples from the test set to generate complaints. This set of complaints is used to compute the loss of a given state at a given time step.

*Benchmark and Results:* We compare our Algorithm from Figure **??** with an offline optimal solution that has been computed to find the state with the minimum average loss over the arrival sequence of complaints. The results are averaged over 10 independent runs.

The results are shown in Figure 11 and Figure 12. We show results for two values of the threshold parameters and in each case plot the loss of the algorithm as compared to the benchmark, as well as the states chosen by the algorithm, as a function of the number of time steps. As can be seen from Figure 11 our algorithm quickly converges to the offline optimal solution after an initial exploration

phase. To get a better understanding of the performance of the algorithm in the initial phases, in Figure 12 we plot the same setting as in the case of Figure 11, but with $x$-axis on a log-scale. For the case of threshold being 0.01, one can see that the state 0 results in much higher loss and, during exploration, the algorithm alternates in a periodic pattern between states 1 and 3 that have similar loss. A similar pattern holds for the case of the threshold being 0.03. It is important to note that the choice of the loss functions was important in this case and that we did not weight each criterion by the volume of the complaints. This demonstrates that our algorithms, when combined with a good choice of the loss function, can be useful in practice.

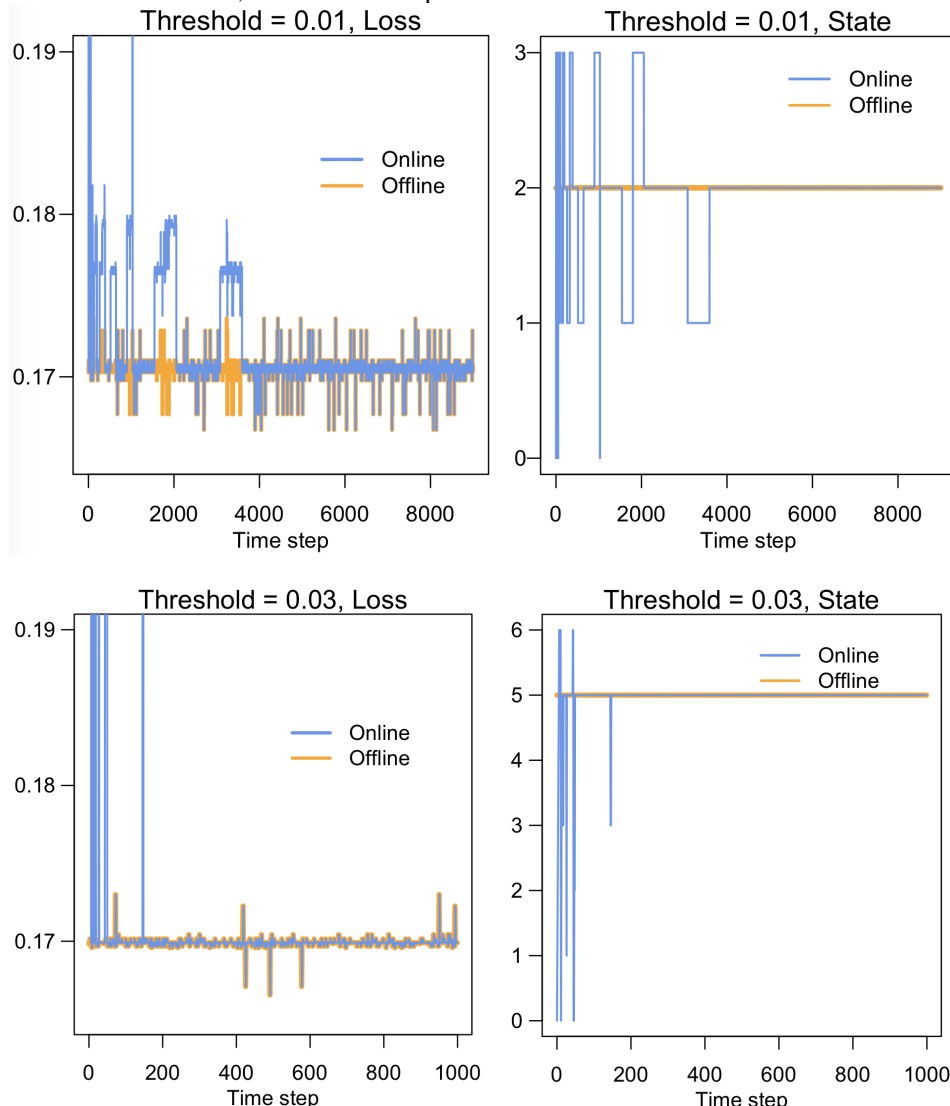

Figure 11: The figure shows the performance of the Algorithm in Figure **??** on the UCI Adult dataset. We present results for two threshold values, and in each case plot the loss of the offline solution and the online algorithm as well as the states chosen by the online algorithm, as a function of the time steps.

**Compute Resources.**    All our experiments were performed on a machine containing a Tesla P100 GPU with 80 GB of RAM and four CPUs.

**Hyperparameters.**    For the case of simulated data the hyperparameters have been mentioned in Section D.1. For the case of real data, apart from the hyperparameters mentioned in Section D.2, we used the default learning rates and optimizers provided by the tensorflow constrained optimization toolkit [Cotter et al., 2018a,b]. We performed a random train/test split as detailed in Section D.2.

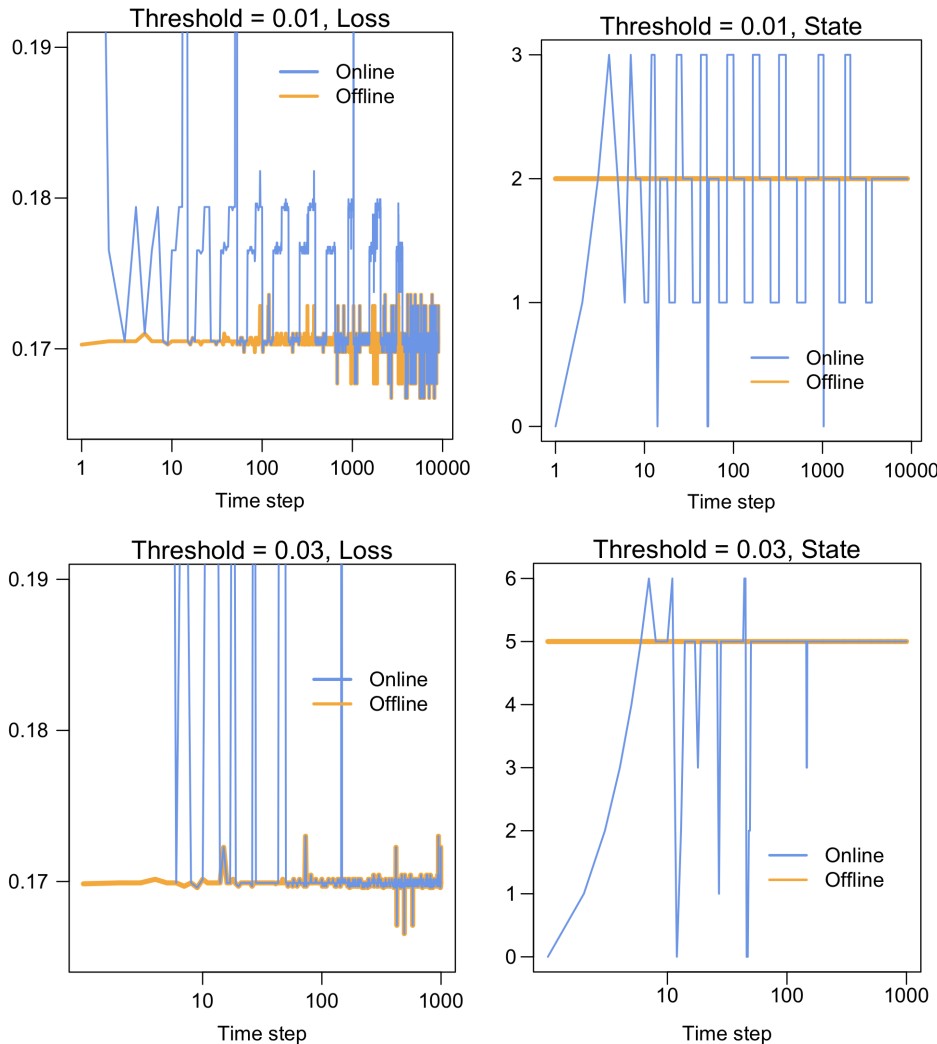

Figure 12: The figure shows the performance ($x$-axis on a log scale) of the Algorithm in Figure **??** on the UCI Adult dataset. We present results for two threshold values, and in each case plot the loss of the offline solution and the online algorithm as well as the states chosen by the online algorithm, as a function of the time steps.

**Assets.** We used publicly available code from the tensorflow constrained optimization toolkit[1] and the publicly available UCI Adult Dataset[2].

# E  Further discussion on the COMPAS example

Throughout the main sections, we have mentioned that the choice of the loss function is important in the effectiveness of our model. We briefly discussed this in Section 3. Below, we present a more detailed discussion of the effect of the loss function on our model, by using the COMPAS scenario from Section 1 as an example.

*Loss function – COMPAS illustration.* Consider the COMPAS example with a graph $\mathcal{G}$ with four criteria namely, false positive rate on population $A$, false positive rate on population $B$, AUC score for

---

[1]License at: https://github.com/google-research/tensorflow_constrained_optimization/blob/master/README.md.

[2]https://archive.ics.uci.edu/ml/datasets/adult.

population $A$ and AUC score for population $B$. We want to understand what kinds of loss functions will result in an overall suboptimal system when our model and algorithms from Section 3. Suppose our algorithm take an action to fix a criterion and reach a state where the true positive rates and the AUC scores associated with the four criteria are: $[0.1, 0.8, 0.5, 0.5]$. Then a poor choice of the loss function would be $f_1 \cdot 0.1 + f_2 \cdot 0.8 + f_3 \cdot 0.5 + f_4 \cdot 0.5$, where $f_i$ represents the fraction of complaints that trigger criterion $i$. Such a choice of the loss function will make our system vulnerable to the loudest voices in the system and as a result might not lead to a good solution at all. A more reasonable choice of the loss is $0.1 + 0.8 + 0.5 + 0.5$, that weighs each criteria equally and does not take into account the underlying size of the population. Another alternative is $\lambda_1 |0.1 - 0.8| + \lambda_2 (|0.5 - 0.5|)$, that aims at keeping both the discrepancy in the false positive rate and the AUC scores small. Finally, the choice we make in our experiments of penalizing each criterion for the deviation from the value over the entire population, i.e., $\max(0, |tpr_{overall} - tpr_w| - \tau)$, also leads to good solutions empirically.

Another case where additive losses are a poor choice is if the criteria in $\mathcal{G}$ is not chosen carefully. For instance, consider a scenario in the COMPAS example where all except one of the criteria correspond to the performance of the system on population $A$. An additive loss would then naturally force the system to disproportionately favor population $A$ over a period of time.

While the above discussion used the COMPAS scenario as a specific example, we would like to re-iterate that our model and algorithms are much more general and can be motivated from different applications. As another motivating scenario for our work, consider a large organization that is building a classifier to detect harmful content that the users of their platform may be exposed to. The organization wants to build a classifier that has a good overall performance, say measured in terms of false positive rates (FPRs) and false negative rates (FNRs) (these in general could be arbitrary metrics). Furthermore, the organization also wants to ensure good FPRs and FNRs on users sliced by different attributes such as race, gender, geographic location, education level etc. While the overall classifier performance is still of paramount importance, the organization's policy team may have given them guidelines to try and enforce that FPRs and FNRs on different slices are less than a certain threshold. However, not all such constraints may be satisfiable and the organization wants to figure out the optimal tradeoffs between these metrics via end user feedback. Our model and algorithms address this question

