# OpenReview forum: "A Theory of Learning with Competing Objectives and User Feedback"
_NeurIPS.cc/2022/Workshop/TSRML — TSRML2022_

### Official Review · Reviewer_UpVf · 2022-10-22

**Overall Rating:** 6

**Summary:**

This paper studies conflicting objectives in a system and how to use user feedback to dynamically improve the system.

**Strengths:**

* This paper studies an interesting problem on complicated system with conflicting objectives
* The theoretical analysis is thorough

**Weaknesses:**

* The definition of the adversarial setting is unclear. On line 250 the paper defines the adversarial setting as " at each time step, multiple complaints arrive for the vertices in graph". But why this can be an adversarial setting, which means the worst case setting, is unclear. Also it is unclear that how this paper is related to machine learning safety.

**Overall Recommendation:**

Overall I think this paper might be a good paper, which studies an interesting problem with thorough theoretical analysis. But the definition about adversarial setting is unclear and it is also unclear that how this paper is related to machine learning safety.

**Review Confidence:**

2: The reviewer is willing to defend the evaluation, but it is quite likely that the reviewer did not understand central parts of the paper

---

### Official Review · Reviewer_egvt · 2022-10-23
**A nice theory paper combining multi-objective optimization and online learning**

**Overall Rating:** 7

**Summary:**

This paper studies a theoretical formulation faced when deploying machine learning systems in the real world, namely, the need to update the system to account for competing objectives and according to user feedback. The authors first motivate the problem by discussing the practical concerns in this setting and using these to derive a simplified theoretical framework. The authors then present two algorithms for the stochastic setting of their framework, one "explore then commit" style algorithm obtaining $T^{2/3}$ regret and one apparently UCB-style algorithm obtaining $T^{1/2}$ regret. For the adversarial setting of their framework, the authors propose an algorithm with constant competitive ratio by modifying the classical ski rental algorithm to account for the problem constraints. They also conduct simulation experiments using both synthetic and real-world base datasets to validate their theory.

**Strengths:**

The discussion of the motivating practical problem is thorough, and the motivating problem itself is clearly important. They explained their theoretical assumptions and framework clearly, and their results are extensive. There is also a discussion of the limitations and the future work that will be necessary to make this more practical, the most important of which (in my opinion) is the relaxation of the binary restriction on the constraints.

**Weaknesses:**

The authors frame their method as a data-driven approach which can adapt to previously unaccounted-for metrics, but two of the central quantities (the constraints graph and the correlation sets) are assumed to be known. In particular for the constraints graph, one must know all of the possible metrics that may be considered in the future to construct the graph. Also, determining the edges in this graph seems like a challenging problem in its own right. (The authors do make some remarks on how these can be determined from historical data.) Regarding the correlation sets, it seems that these are necessary to avoid searching the exponentially large state space, but it is unclear how one would determine these sets in practice.

Finally, the authors themselves mention this, but the fact that the constraints are binary seems unrealistic. Indeed, for many real-world cases (such as false positive rate vs. AUC for the COMPAS example), it seems like these metrics should actually be *positively* correlated up to a point--for instance, a completely useless model will perform poorly with respect to both metrics, while any reasonable model should improve both metrics above random--, and beyond this point, there is tension between the two metrics. It seems difficult to capture this complex relationship with the binary constraints. So I agree with the authors that this is an important step for future work.

**Overall Recommendation:**

This is a nice theory paper which can provide insight into an important practical problem. I recommend accept.

**Review Confidence:**

3: The reviewer is fairly confident that the evaluation is correct

---

### Decision · Program_Chairs · 2022-10-23

**Decision:**

Accept

**Comment:**

Following the unanimous recommendations from reviewers, the submission is accepted.